

# Three different glacier surges at a spot:
# What satellites observe and what not

*Frank Paul[1], Livia Piermattei[2], Desiree Treichler[2], Lin Gilbert[3], Luc Girod[2], Andreas Kääb[2], Ludivine Libert[4], Thomas Nagler[4], Tazio Strozzi[5], Jan Wuite[4]*

Department of Geography, University of Zurich, 8057 Zurich, Switzerland
Department of Geosciences, University of Oslo, P.O. Box 1047, 0316 Oslo, Norway
UCL-MSSL, Department of Space and Climate Physics, Mullard Space Science Laboratory, Holmbury St Mary, Surrey RH5 6NT, UK
ENVEO IT GmbH, Fürstenweg 176, 6020 Innsbruck, Austria
Gamma Remote Sensing, 3073 Gümligen, Switzerland

Corresponding author: Frank Paul (frank.paul@geo.uzh.ch)

## Abstract

In the Karakoram, dozens of glacier surges occurred in the past two decades, making the region one of its global hotspots. Detailed analyses of dense time series from optical and radar satellite images revealed a wide range of surge behaviour in this region: from slow advances longer than a decade at low flow velocities to short, pulse-like advances over one or two years with high velocities. In this study, we present an analysis of three currently surging glaciers in the central Karakoram: North and South Chongtar Glaciers and an unnamed glacier referred to as NN9. All three glaciers flow towards the same region but differ strongly in surge behaviour. A full suite of satellite sensors and digital elevation models (DEMs) from different sources are used to (a) obtain comprehensive information about the evolution of the surges from 2000 to 2021 and (b) to compare and evaluate capabilities and limitations of the different satellite sensors for monitoring relatively small glaciers in steep terrain. A strongly contrasting evolution of advance rates and flow velocities is found, though the elevation change pattern is more similar. For example, South Chongtar Glacier had short-lived advance rates above 10 km y$^{-1}$, velocities up to 30 m d$^{-1}$ and surface elevations increased by 200 m. In contrast, the neighbouring and three times smaller North Chongtar Glacier had a slow and near linear increase of advance rates (up to 500 m y$^{-1}$), flow velocities below 1 m d$^{-1}$ and elevation increases up to 100 m. The even smaller glacier NN9 changed from a slow advance to a full surge within a year, reaching advance rates higher than 1 km y$^{-1}$. It



seems that, despite a similar climatic setting, different surge mechanisms are at play and a transi-
tion from one mechanism to another can occur during a single surge. The sensor inter-comparison
revealed a high agreement across sensors for deriving flow velocities, but limitations are found on
small and narrow glaciers in steep terrain, in particular for Sentinel-1. All investigated DEMs have
the required accuracy to clearly show the volume changes during the surges and elevations from
ICESat-2 ATL06 data fit neatly. We conclude that the available satellite data allow for a compre-
hensive observation of glacier surges from space when combining different sensors to determine
the temporal evolution of length, elevation and velocity changes.

## 1.  Introduction

Glacier surges in the Karakoram are widespread (e.g. Sevestre and Benn, 2015) and have been
thoroughly documented using historic literature sources and time series of satellite images (Cop-
land et al., 2011; Bhambri et al., 2017; Paul, 2020). A large number of publications provides in-
sights into decadal elevation changes (e.g. Bolch et al., 2017; Berthier and Brun, 2019; Brun et al.,
2017; Gardelle et al., 2013; Rankl and Braun, 2016; Zhou et al., 2017) and mean annual flow ve-
locities (e.g. Dehecq et al., 2015; Rankl et al., 2014) at a regional scale. Using various satellite da-
tasets, several studies have also investigated individual glacier surges at high temporal resolution
(e.g. Bhambri et al., 2020; Mayer et al., 2011; Paul et al., 2017; Quincey et al., 2015; Round et al.,
2017; Steiner et al., 2018).

This increasing interest is in part due to the hazard potential of glacier surges, in particular when
river damming creates lakes that might catastrophically drain in so-called glacier lake outburst
floods (GLOFs) with far reaching impacts (e.g. Bazai et al., 2021; Bhambri et al., 2019 and refer-
ences therein; Iturrizaga, 2005), but also due to the increased availability of satellite data for char-
acterizing surges in detail (e.g. Dunse et al., 2015; King et al., 2021; Nuth et al., 2019; Rashid et
al., 2020; Wang et al., 2021; Willis et al., 2018). The still limited understanding of surges in the
Karakoram region (e.g. Farinotti et al., 2020) and the high diversity of observed surge characteris-
tics (e.g. Bhambri et al., 2017; Hewitt, 2007; Paul, 2015; Quincey at al., 2015) also contribute to
the recent efforts. These studies found that both main types of glacier surges can be found in the
Karakoram, sometimes side-by-side: The Alaska type, which might be triggered by a change in the
basal hydrologic regime, creates pulse-like surges of a short duration (2-3 years), whereas the
thermally-controlled Svalbard type has often active surge durations of many years (e.g. Jiskoot,
2011; Murray et al., 2002; Raymond, 1987; Sharp, 1988). Although the physical reasons for the
differences and variability of surges in the Karakoram are yet unknown (e.g. glacier properties,
thermal regime, mass balance history), many glaciers in the Karakoram have surged repeatedly,
sometimes at surprisingly constant intervals and over centuries (e.g. Bhambri et al., 2017; Paul,



2020). On average, surges in the central Karakoram repeat after 40 to 60 years, but intervals can
range from less than 20 to more than 80 years.

In the thermally controlled case, it is sometimes difficult to distinguish a regular advance from a
surge, as the transition can be gradual (Lv et al., 2020). Whether an advance (stimulated by a posi-
tive mass budget) is indeed a surge might be determined by comparison with the behaviour of
neighbouring glaciers. As thresholds on advance rates or ice flow speedup might not be efficient to
distinguish (slow) surges from advances in the Karakoram, the typical mass redistribution pattern
of a surge (from an upper reservoir to a lower receiving zone) as obtained from differencing digital
elevation models (DEMs) acquired a few years apart (e.g. Gardelle et al., 2013) is a more reliable
identifier (Lv et al., 2019; Goerlich et al., 2020). Usually, the surface in the upper regions of a
glacier does not lower significantly during a regular advance (Lv et al., 2020). A further method to
discriminate surges from a usual advance is related to a strong increase in crevassing and devel-
opment of shear margins. However, these are only visible in very high-resolution satellite images
or time-series of SAR data (Leclercq et al., 2021).

In this study, we present (a) a comparative analysis of the on-going surges of three glaciers in the
central Karakoram: North and South Chongtar Glacier and a small, unnamed glacier referred to
here as NN9. We present a comparative analysis of their changes in length, advance rates, flow
velocities and surface elevations to elucidate the respective similarities and differences in surge
behaviour. As a second aim of this study, we (b) investigate the feasibility of various satellite sen-
sors and DEMs to follow the temporal evolution of the surges comprehensively. Included are opti-
cal (Sentinel-2, Landsat, Planet cubesats) and synthetic aperture radar (SAR) imaging sensors
(Sentinel-1, TerraSAR-X), altimeter data from ICESat-2 and DEMs from the Shuttle Radar To-
pography Mission (SRTM), the Satellite Pour l'Observation de la Terre (SPOT), the High Moun-
tain Asia DEM (HMA-DEM) and the Advanced Spaceborne Thermal Emission and reflectance
Radiometer (ASTER) by Hugonnet et al. (2021).


## 2.  Study region

The study region is located in the central Karakoram, north of the Baltoro Glacier, at about 5.94°
N and 76.33° E (Fig. 1). East of the study region stands the second highest mountain in the world,
the 8611 m high K2. Slopes of the surrounding terrain are very steep and snow avalanches from
the surrounding rock walls are a major source of glacier nourishment. Mass changes over the past
20 years derived from satellite data using the geodetic method show more or less constant near-
zero mass budgets in the study region (Hugonnet et al., 2021), confirming the continuation of the
'Karakoram Anomaly' (i.e. the balanced mass budgets) in this region (Farinotti et al., 2020).



*Fig. 1: Overview study region*

Most precipitation in the study region is brought by westerly air flow during winter, but the mon-
soon brings moist air from the southeast also during summer (Maussion et al., 2014), falling as
snow at the high elevations of the rock walls surrounding most glaciers. However, due to the good
protection from nearly all directions, the amount of snowfall in the study region is limited and a
dry-continental climate can be expected (e.g. Sakai et al., 2015). As surge-type glaciers are abun-
dant (Copland et al., 2011; Bhambri et al., 2017) and repeat intervals are comparably short (Paul,
2020), several glaciers in the Karakoram are typically actively surging at any given time.

The three glaciers investigated here (North/South Chongtar, NN9) have mean elevations around
5500 m and are surrounded by mountain ridges with elevations between 6000 and 7500 m above
sea level. South Chongtar Glacier (shortened to South Chongtar in the following) is the largest
with an area of ~31 km$^2$ and a length of more than 14 km at minimum extent, but it has a narrow
tongue with a near-constant width of about 800 m. The glacier is mainly east-west oriented in its
upper part, bending towards south-north near the terminus. North Chongtar lies north of South
Chongtar and is connected to it in its accumulation area. It flows from southeast to northwest, co-
vers an area of ~10 km$^2$, has a length of 4.5 km at minimum extent and is about 400 m wide. The
unnamed glacier NN9 is located on the opposite side of the main valley and flows roughly from
west to east. The glacier is about 3.5 km long at minimum extent with an area of 4 km$^2$ and a ~300
m wide tongue. Table 1 summarises further characteristics and topographic properties.

*Table 1: Basic properties of the three investigated glaciers*


At their maximum extent the three glaciers reach Sarpo Laggo Glacier, a compound-basin valley
glacier with a size of 122.3 km$^2$. This glacier experienced a massive surge shortly before 1960
(Paul, 2020) and a smaller, more internal one (i.e. not reaching the terminus), between 1993 and
1995 (e.g. Paul, 2015). According to Paul (2020), South Chongtar had a rapid advance during a
surge that started in 1966 with a short active phase of about two years followed by a quiescent
phase with continuous down-wasting and retreat. During this surge it partly compressed the ice
from Sarpo Laggo and deformed a moraine from Moni Glacier (see Fig. 1), leaving an impressive
surge mark. In contrast, North Chongtar started advancing about 55 years ago but has not yet
reached Sarpo Laggo. The Shipton map from 1937 (Shipton, 1938) shows North Chongtar in con-
tact with it, indicating that the terminus might reach it again. The glacier NN9 had its last surge
from about 1961 to 1971 (leaving a small surge mark on Sarpo Laggo) and retreated in its quies-
cent phase until 2000, when it started to advance slowly. The two glaciers to the south of NN9
(NN7 and NN8 in Paul, 2020) both surged around 1955, and again in 1998 and 1980, respectively.





NN8 also surged after 2002, indicating a surge cycle of only 20-25 years. The next surge of NN8
can thus be expected soon, at least if environmental conditions prevail.


## 3.   Datasets

In this section, we describe the satellite and auxiliary datasets used to derive time series of glacier
outlines, surface flow velocities and elevation changes in the study region. Figure 2 shows the
temporal coverage of each dataset, and the periods selected for the analysis. Changes in glacier
extent have been mapped for the active (advance) phases of the three glaciers, starting with Land-
sat Multispectral Scanner (MSS) images from 1973 for North Chongtar Glacier. The earliest da-
tasets used to derive flow velocities and elevation changes were acquired in 2000, the Landsat 7
Enhanced Thematic Mapper plus (ETM+) panchromatic band and the SRTM DEM.

*Fig. 2: Timeline of datasets used*

### 3.1 Glacier extent and centrelines
We used glacier outlines from the updated Glacier Area Mapping for Discharge from the Asian
Mountains (GAMDAM2) inventory by Sakai (2019) as a starting point for all glacier extents. This
dataset was locally improved (removing rock outcrops and seasonal snow) using a Landsat 8 im-
age acquired on 21 October 2020 (Fig. 1). Given the unknown final length of the glaciers, we dig-
itized possible virtual maximum extents for the three glaciers, avoiding overlapping polygons in
their terminus regions. The virtual extents were guided by maximum extents of previous surges
described by Paul (2020).

Changes in extent were derived from time series of spatially consistent Landsat data (MSS, TM,
ETM+ and Operational Land Imager (OLI)) from path-row 148-35 as well as a couple of Sentinel-
2 scenes (tile 43SFV) required to bridge sparse cloud-free Landsat scenes after February 2021. The
list of satellite scenes used for determination of geometric changes (outlines, length changes) is
given in Table S1 of the Supplemental Material.

The centrelines for NN9, South and North Chongtar were manually digitized starting from the
highest points of each glacier down to the virtual maximum extent. The centrelines were divided
into equidistant points of 100 m at which values for velocity and elevation were extracted.

### 3.2. Flow velocity
Time series of optical and SAR data were used to derive glacier flow fields (listed in Table S2).
Landsat 7 and 8 scenes, Sentinel-2 and TerraSAR-X (TSX) were used (Fig. 2) to determine pre-



surge flow velocities of South Chongtar and advance/surge phase velocities for all glaciers. Images
from Planet cubesats were used for a comparison of results with Sentinel-2 and some gap filling in
the time series rather than for a full documentation of the active surge of South Chongtar. Veloci-
ties from Landsat 8 were also compared with Sentinel-2.

From TSX co-registered single-look slant range complex (SSC) images acquired in StripMap
mode, with across- and along-track resolution of up to 3 m are used. The selected image pairs are
from two different tracks, cover the study region in the descending direction and were acquired in
2011, 2012 and 2014 (Table S2). Time series of Sentinel-1 single-look complex (SLC) data ac-
quired in interferometric wide (IW) swath mode were used to test its feasibility to derive flow ve-
locities and to create an animation of the surge that is unobstructed by clouds. The Sentinel-1 IW
SLC data have a nominal ground resolution of 5 m x 20 m.

**3.3 Elevation information**
To follow elevation changes of the glaciers before and during the surge, we analysed several
DEMs from both optical and SAR sensors (Table 2). We used the following DEMs with known
acquisition dates: The SRTM1 DEM at 1 arcsec (~30 m) resolution from February 2000 (USGS,
2017), a SPOT5-HRS DEM from January 2010 (Gardelle et al., 2013; we used their version v2 for
rugged areas), a SPOT6 DEM from October 2015 (Berthier and Brun, 2019), and a SPOT7-
derived DEM from October 2020 that was generated for this study. In addition, we used the HMA-
DEM mosaic (Shean, 2017) as a reference for DEM co-registration analysis due to its superior
spatial resolution and accuracy over stable terrain (off-glacier) compared to the other DEMs (Fig.
S1). The HMA-DEM is composed of various DEM datasets mostly acquired during 2015 (Feb.,
April, July, and Aug.) in this region. Elevation values along the centrelines are extracted from the-
se DEMs and DEM differences are calculated for the periods 2000-2010, 2010-2015 and 2015-
2020. For comparison, we also analysed elevation changes derived from ASTER time series by
Hugonnet et al. (2021). These provide additional information about the 2000-2005 and 2005-2010
period as well as from 2000 to 2019 (before the surge of South Chongtar).

*Table 2: Overview DEM characteristics*


We also analysed whether altimetry data from ICESat-2 could be used to reveal elevation changes
at a higher temporal resolution. The Advanced Topographic Laser Altimeter System (ATLAS)
instrument on-board ICESat-2 acquires elevation profiles at a 91-day temporal resolution since
October 2018. Each satellite overpass results in three beam pairs that are separated by 3.3 km and
90 m between/within pairs, respectively (Markus et al., 2017). The ICESat-2 ATL06 dataset pro-
vides geolocated land ice surface heights with 40 m spatial resolution in profile direction. Figure
S2 shows the ATL06 dates and elevations of data points crossing North and South Chongtar, and



the two closest repeating pairs of tracks on South Chongtar. Due to the systematic off-pointing at
mid-latitudes, ICESat-2 tracks are not repeated exactly in our study area and the ATL06 data alone
proved too sparse, both geographically and temporally, for further analysis of the surges.

The ICESat-2 ATL03 Global Geolocated Photon Data (Neumann et al., 2021), from which the
ATL06 dataset is a higher-level derivative, provides surface elevation measurements from individ-
ual photons every 0.7 m along the elevation profiles, revealing details of the surface topography of
the glaciers. The ICESat-2 surface elevations fall into the time gap of the DEMs between 2015 and
2020, thus providing additional temporal information on the surge development. In total, we found
42 intersections with the centrelines of the three investigated glaciers: 23 on South Chongtar (from
seven dates), 13 on North Chongtar (from six dates), and 6 on NN9 (from three dates).


## 241 4. Methods

### 242 4.1 Glacier extent

The timing of the selected images used to digitize glacier extents varies strongly depending on the
advance rates. To have at least a two-pixel change in frontal position, it varies from several years
for the slow advance of North Chongtar to about 16 days for the surge phase of South Chongtar.
Due to frequent cloud cover, different scenes had to be used for the individual glaciers (Table S1).
For the digitization, the polygon referring to the virtual maximum extent of each glacier was split
into a multi-polygon by digitizing the smaller extents visible on the respective satellite images.

Length changes between two terminus positions from $t_1$ and $t_2$ were derived manually using the
distance tool in ArcGIS. Several values were obtained for each change and a suitable average as-
signed (values usually varied by about ±10 m). We only used the Landsat 7 and 8 time series for
this as the Landsat Collection 1 data had a spatial shift compared to Sentinel-2 (e.g. Paul et al.,
2016). The length change values from $t_1$ to $t_2$ were divided by the temporal difference $(t_2 - t_1)$, con-
verted to mean annual advance rates and assigned to the date that is halfway between $t_1$ and $t_2$.
Cumulative changes were obtained by summing up the individual length changes.

### 258 4.2 Velocities

Flow velocities typically span two to three orders of magnitude, e.g. from <0.1 m d$^{-1}$ for near stag-
nant glaciers to >10 m d$^{-1}$ during a surge. When using offset-tracking (e.g. Strozzi et al., 2002) this
range can to some extent be accounted for by varying the search window size or the time between
the acquisition dates of the image pair. If glaciers with very different flow velocities are in the
study region, it might be required to use images from different dates for the analysis or an adaptive
search window (Debella-Gilo and Kääb, 2012). In the following, we describe some basics of the
processing lines applied for optical and SAR sensors.






The normalized cross-correlation algorithm implemented in the correlation image analysis soft-
ware (CIAS, Kääb and Vollmer, 2000) is used to calculate the glacier surface displacement be-
tween optical satellite image pairs (Fig. S3 is illustrating the workflow). The satellite images were
not co-registered as we assume that they are corrected for topographic distortion, and therefore the
displacement calculated between two images is the actual horizontal displacement without any
influence of topography. To check co-registration, abundant stable terrain was included in the cor-
relation. The displacements are estimated at a spatial resolution of 100 m while the size of the
search area is set in relation to the maximum displacement estimated between two satellite scenes.
Dividing the displacement by the temporal difference between the image pairs (Table S2) gives
velocity in m d$^{-1}$.

With optical data, clouds, cast shadows and changes in snow cover lead to false detections or bi-
ased measurements of the calculated displacement fields. These mismatches are removed in post-
processing by setting a threshold of the maximum correlation coefficient (<0.5) and velocity. For
Sentinel-2 data, elevated objects such as clouds are detected by applying CIAS between band 4
and band 8 of the same Sentinel-2 scene. The calculated perspective displacements (both bands are
recorded at slightly different positions of the sensor) are then used to mask the clouds. For all sat-
ellite data, spatial filtering based on a moving median window as well as temporal filtering are ap-
plied to remove additional outliers and noise (Fig. S3).

Surface flow velocities for TSX data were derived by an iterative offset-tracking technique devel-
oped for SAR data (Wuite et al., 2015). This method does not require coherence and is thus also
capable of acquiring flow velocity data over longer time spans and in regions with fast flow. The
method is based on cross-correlation of templates in SAR amplitude images and provides both the
along-track and line-of-sight velocity components from a single image pair. We used a template
size of 96×96 pixels for generating velocity maps with 50 m grid spacing and applied a 9x9 in-
verse-distance median filter in the post-processing step to remove outliers and fill in small gaps.
For Sentinel-1, the same method was applied but tests with various image template sizes were per-
formed with an image pair acquired on 4 and 1611. 2020 during the peak of the surge (Fig. S4).

**4.3 Elevation data**
We used the MicMac software to generate the SPOT 2020 DEM from the raw imagery (Rupnik et
al., 2017). The pre-processing of all DEMs follows the standard processing steps for DEM differ-
encing. All DEMs were projected to UTM 43N (EPSG 32643), elevations were vertically trans-
formed to the WGS 84 ellipsoid and DEMs were co-registered to the HMA DEM using OPALS
(Pfeifer et al., 2014). Specifically, we applied least squares matching to estimate the full 3D affine
transformation parameters that minimize the errors with respect to the reference DEM over com-



mon stable areas. These were manually digitized off-glacier excluding slope values larger than 40
degrees (Fig. S5). We also excluded data voids from further analyses. This removes large parts of
the accumulation areas of some glaciers in the case of the SPOT 2010 and 2015 DEMs, but had
little impact for the other DEMs.

All DEMs were resampled, clipped and aligned to the same 30 m grid and a high-resolution 5 m
grid for the HMA DEM and SPOT 2020. We did not correct the SRTM DEM for microwave pene-
tration into ice and snow (Gardelle et al., 2012), as the effect is small and uncertain compared to
the elevation differences caused by the surges. Elevation values were extracted along the centre-
lines and subtracted from SRTM.

To estimate volume changes resulting from the surges, volume gain and loss (i.e., summing up all
positive and negative values within the tongues) were calculated for each glacier tongue with ad-
justed extents and glacier-specific epochs (Fig. S5). For comparison, we included the glaciers NN7
and NN8 (see Fig. 1) in the analysis as they also surged during the study period.

As the ICESat-2 ATL06 datasets did not provide useful results, only the ATL03 dataset was fur-
ther processed using python libraries geopandas (Jordahl et al., 2021), rasterio (Gillies et al., 2021)
and shapely (Gillies et al., 2021a). The photon elevations were filtered to only retain elevation
samples      classified      as      likely      land      or      ice      surfaces      (parameters      sig-
nal_conf_ph/signal_conf_ph_landice >1) and classified into glacier and off-glacier samples using
maximum glacier outlines. On the bright glacier surface, both the weak and strong laser beams
yield sufficient photon returns for complete elevation profiles. This is less true for moraines /
rocky areas (profile 3 in Fig. S6), where the weak beam yields considerably fewer surface returns.

Elevation values were sampled for all elevation points (containing a DEM cell) and the AMES
stereo pipeline (version 2.7.0, Shean et al., 2016) was used to co-register the elevation profiles (on-
ly off-glacier samples) with the already co-registered DEMs used in this study (no co-registration
offset was found). The profiles were intersected with the glacier centrelines to compare the ATL03
elevation samples with the DEMs. The median of all elevation samples on each profile within a 10
m buffer from the centreline are used as surface elevations at the intersection points.

**4.4 Uncertainties**
The uncertainty of the length change data has been determined by measuring for each glacier and
each time step different points at the terminus. From the range of values, a reasonable mean value
was determined manually. Glacier terminus positions were digitized only once and used only for a
qualitative illustration (outline overlay) of the changes, i.e. we have not explicitly calculated un-
certainties of glacier extents. As a range of sensors with different spatial resolutions is used for the



digitizing (e.g. Landsat MSS, ETM+, OLI and Sentinel-2), the uncertainty varies with the sensor.

Based on the assumption that measurement errors over glaciers and other terrain are common
(Paul et al., 2017), we assessed the uncertainties of glacier flow velocities from stable terrain ve-
locity observations, where flow velocities are supposed to be zero, using the same stable areas as
used for DEM co-registration (Fig. S5). Uncertainties are derived as measures of median and a
robust standard deviation based on the median absolute deviation (MAD), which is a bit less sensi-
tive to outliers (e.g. Dehecq et al., 2015). Co-registration accuracy of the DEMs was computed
from elevation differences calculated over stable terrain (off glacier) with slopes smaller than 40°
(Fig. S5).


**5.   Results**

**5.1 Changes in glacier extent and morphology**
In Fig. 3 the temporal evolution of terminus positions is depicted as an overlay of extents showing
slow advances of NN9 (starting in 2000) and North Chongtar (since 1973) along with a rapid ad-
vance of South Chongtar (starting mid-2020). For better visibility, the retreat phase of South
Chongtar from 2000 to mid-2020 is not shown. Snapshots of the geometric evolution can be found
in Fig. S7 for the time period before the surge of South Chongtar (1993-2019) and in Fig. S8 for
the time during its surge (2020-2021). The related cumulative length changes for all three glaciers
are shown in Fig. 4a for their advance phases, whereas Fig. 4b only shows advance rates for the
glaciers NN9 and North Chongtar (they were out of scale for South Chongtar).

*Fig. 3: Multi-temporal outline overlay advance phase*


**South Chongtar** entered its quiescent phase after its 1966/67 surge and exhibited constant thin-
ning with limited frontal retreat over several decades. After 30 years (in 2000) the former surge
lobe was still largely ice filled, though increasingly debris covered. Driven by further thinning, a
clear retreat of the terminus (remaining clean ice) became visible after 2000, reaching about -800
m by 2009 and -2300 m by mid-2020. During this retreat phase, its middle part always showed
some residual flow, i.e. it was not completely stagnant. In 1993 a deformation of the medial mo-
raine started moving forward, about 300 m by 2009 and 500 m by 2019.

In 2017, a new surge developed with the typical funnel-shaped appearance of the front. While the
lowest part of the glacier was still thinning and retreating in 2019, the surge front reached the ter-
minus in July 2020 and the front started advancing by about 3 km in 10 months (Fig. 4a) with ad-
vance rates of up to 12.6 km yr$^{-1}$ (35 m d$^{-1}$) in early Nov. 2020. During this time the lower part





widened massively and the entire surface became heavily crevassed. The front advanced into its
former surge mark on Sarpo Laggo Glacier and pushed the ice surrounding it towards the opposite
side of the valley. By June 2021 advance rates decreased considerably, but the terminus was still
advancing.

*Fig. 4: Cumulative length changes and advance rates*

**North Chongtar** on the other hand, advanced at a more or less constant rate of 30 m y$^{-1}$ until 2004
when it passed a total of 800 m since 1973 (earliest MSS image, see Table S1). A very high-
resolution satellite image from 2001 is available in Google Earth and shows some crevassing near
the terminus but not a surging glacier. There is no indication of a melt water stream leaving the
glacier snout. After 2005, advance rates increased linearly and we assign this as the onset of the
surge phase. This increase resulted in a nearly completely crevassed surface and widespread shear
margins. Both are also visible in the 15 m resolution Landsat panchromatic bands, and, even bet-
ter, in very high-resolution images from 2011 and 2016 available in Google Earth. In 2013 the
terminus reached a step in the valley slope, creating a deep transverse crevasse that seemed to sep-
arate the lowest part of the tongue but actually didn't. By 2021 nearly the entire surface was still
crevassed and the glacier had advanced by a further 1600 m since 2004, i.e. 2.4 km in total.

The small valley **glacier NN9** slowly retreated until 1998 and started advancing a year later at
about a constant rate of 40 m y$^{-1}$ until 2016 (Fig. 4). Up to this year, its lowest parts had some cre-
vasses but looked otherwise like a usual advancing glacier. This changed a year later when the
glacier thickened considerably, developed shear margins and started advancing at a much higher
rate of up to 1000 m y$^{-1}$ in 2021, indicating the start of the surge phase. The increasingly crevassed
surface also became visible in Sentinel-2 images and with the 15 m Landsat 8 band. The total ad-
vance from 1999 to 2018 was 800 m followed by a further 500 m until June 2021. In July the
frontal advance accelerated further reaching nearly 3 km y$^{-1}$ in August 2021, whereby the lower
part of the tongue separated from the main glacier and slid down the remaining kilometre in about
a month. More ice is following from higher elevations, possibly leading to some interaction with
the still advancing terminus of South Chongtar.

**5.2 Flow velocities**
**5.2.1 NN9 and North Chongtar**
Selected flow velocity maps for the two glaciers are shown in Fig. 5 and related velocity profiles
along the centreline of the main trunk can be seen in Figs. 6a and b for NN9 and North Chongtar,
respectively. The c. 300-400 m wide tongue of NN9 is at the edge of the possibilities for deriving
flow velocities with offset-tracking (and a 100 m grid) from the optical sensors, but the high reso-
lution of TSX StripMap acquisitions provides near-complete spatial coverage (Fig. 5b). Due to





local cloud cover several of the optical image pairs selected for South Chongtar could not be used
for NN9 and North Chongtar.

Though scattered, the values derived from Landsat 7 (Fig. 5a), Sentinel-2 (Fig. 5c) and Landsat 8
(Fig. 5d) look reasonable. Pre-surge values are around 0.1 m d$^{-1}$ with Landsat 7 (2000-2002) and
TSX (2011 and 2012) and a bit higher (up to 0.2 m d$^{-1}$) with Sentinel-2 in 2017 (Fig. 6a). After-
wards values in the lower part of NN9 (between 2.5 and 4 km) start increasing to 0.4 m d$^{-1}$, reach-
ing 0.8 m d$^{-1}$ between August and October 2020. The upper glacier started accelerating in autumn
2020 with a near-linear increase up to the terminus (Fig. 6a), indicating surge activation in the
lower part of the glacier. The increased crevassing of NN9 is also visible in the higher intensity
values of the Sentinel-1 animation towards the latest images (see Supplemental Material).

*Fig. 5: 2D flow velocity maps 2000-2019 for all glaciers*



For the larger North Chongtar, a slightly better coverage can be obtained from the optical sensors
than for NN9. The most homogenous flow fields are derived by TSX (Fig. 5b) indicating higher
flow velocities of up to 0.4 m d$^{-1}$ in its lower two thirds up to the terminus in 2012. The profiles in
Fig. 6b from Landsat 7 show similar values. Velocities derived from Sentinel-2 between 2016 and
2019 are lower in the region from 3 to 5.5 km. There is a zone with very low velocities between
4.5 and 5 km and acceleration afterwards. From August 2019 to October 2020 flow velocities are
between 0.8 and 1 m d$^{-1}$ near the terminus, indicating that this region is fast flowing and advanc-
ing, whereas the upper regions are still moving with 0.2 to 0.4 m d$^{-1}$.

*Fig. 6: 1D centre-line velocity profiles NN9 & South Chongtar*


**5.2.2 South Chongtar**
The much larger South Chongtar glacier was adequately captured by the optical sensors so that a
more continuous flow field could be derived (Fig. 5) and pre-surge flow evolution could be fol-
lowed in detail (Fig. 7a). Comparing the maps in Fig. 5, a slow but steady increase of flow veloci-
ties from 2000 to mid-2019 over large parts of the glacier can be seen, starting at about 0.15 m d$^{-1}$
and ending at 0.4 m d$^{-1}$. These values are similar to the other two glaciers, but affect a larger re-
gion. The temporal evolution shown in Fig. 7a confirm this observation, pre-surge flow velocities
are highest (up to 0.2 m d$^{-1}$) near the middle of the glacier (around 8 to 10 km) and decrease gradu-
ally to 0 m d$^{-1}$ at its highest and lowest points. In the region between 11 and 14 km the gradual in-
crease of flow velocities can be followed from 2000/02 (with Landsat 7) to 2014 (with TSX).
Mean annual values with Landsat 8 from 2013 to 2014 match perfectly with mean monthly TSX
values from April to May 2014. Landsat 8 velocities from 2013 to 2016 and Sentinel-2 from 2016





to 2019 show the continuation of the slow velocity increase over the entire glacier length, reaching
0.4 m d$^{-1}$ in 2018/19. A direct comparison with Landsat 8 over nearly the same period (grey dots in
Fig. 7a) is shown on top of the curve from Sentinel-2, indicating again a near-perfect match. In
August 2019 the gradual increase changed at first rapidly to 0.8 m d$^{-1}$ and then more slowly to 1.1
m d$^{-1}$ between September 2019 and June 2020. With the stagnant terminus still at 13.5 km, the
strong velocity increase behind the front marks the onset of the surge around August 2019.

*Fig. 7: Velocities South Chongtar*


The last curve from Fig. 7a is repeated in Fig. 7b (dark blue at the bottom), as we had to switch the
scale for better visibility of velocities during the surge phase. Flow velocities increased to about 4
m d$^{-1}$ by July 2020. In August 2020 we could derive detailed flow fields from Sentinel-2 images
acquired only 5 days apart. A sharp surge front with maximum velocities formed, reaching values
of more than 25 m d$^{-1}$ in August/September 2020. With peak velocities near 30 m d$^{-1}$ as derived
locally from Planet imagery (Fig. S9), South Chongtar Glacier had likely one of the highest flow
velocities ever measured in the Karakoram region. Behind this maximum, flow velocities de-
creased about linearly back to 3 km along the centre line. When the surge front reached the termi-
nus in July 2020, a rapid advance started (see 5.1). Velocities dropped to 15 m d$^{-1}$ by November
2020 and below 10 m d$^{-1}$ by January 2021. Afterwards, maximum velocities near km 15 changed
only slowly over the entire glacier length, indicating that the active surge was on-going. Around 10
km along the centre line, velocity is still around 5 m d$^{-1}$ in early May 2021 or 40 times higher than
during the quiescent phase (Fig. 7b). The related Hovmöller diagram for the surge phase in Fig. 7c
confirms the strong pulse-like acceleration in August 2020 with a rapid decline afterwards. The
corresponding 2D plots of flow velocities during the surge phase of South Chongtar (Fig. 8) also
reveal the rapid velocity increase by September 2020 and the decrease afterwards.

*Fig. 8: 2D maps of surface velocities South Chongtar*


The spatial distribution of highest flow velocities of Figs. 8b and c are not symmetric to the centre
line, indicating that the deformation-related maximum flow velocity in the centre of a glacier has
reduced relevance here. This somehow counterintuitive behaviour indicates that during a surge
basal sliding is the process dominating over deformation. Other possibilities are a decreased re-
sistance of the valley floor or because of the topography redirecting the mass flow from northwest
to north. The cross-profile flow velocities (Fig. 9) reveal that this pattern persists throughout the
entire surge.

*Fig. 9: 1D cross-profile velocities South Chongtar (surge phase)*




**5.3 Elevation changes**

In the three panels of Fig. 10 we show differences in elevation between the SRTM DEM and the other four DEMs along the centrelines of the three glaciers. Additionally, differences from selected ICESat-2 ATL03 points are plotted. Figure 11 shows related elevation change maps for 2000 – 2010, 2010 – 2015, 2015 – 2020 and 2000 – 2020. DEM differences obtained from ASTER in an independent study (Hugonnet et al., 2021) have been used for comparison.

The elevation data for **NN9** (Fig. 10a) show virtually no change in its upper part down to 3.5 km where the terminus was located in 2000. The ICESat-2 data adds no further information here, as all available data points are located in the upper part. Below this region, the 'elevation gain' due to the advancing snout can be followed down to km 4.5 in 2020. The small region of elevation gain by the advancing tongue is also visible in each of the maps in Fig. 11. The elevation differences between the two high-resolution DEMs from 2015 and 2020 in Fig. 11c reveal some surface lowering in the upper part, but over the longer period 2000 to 2020 (Fig. 11d) this lowering nearly disappears (i.e. is smaller than the SRTM uncertainty). So for NN9 the typical mass transfer of a surge could not be observed until October 2020 and elevation changes look like expected for a usual advance rather than a surge.

For **North Chongtar** (Fig. 10b) the situation is similar, but a surface lowering of about 40 m can be observed at higher elevation. The SPOT data from October 2015 and ICESat-2 data points from December 2018 at 4.2 km indicate that the largest changes happened between 2015 and 2018. Accordingly, this change is well visible in the high-resolution 2015-2020 DEM difference (Fig. 11c) and the differences over the full 2000-2020 period (Fig. 11d). However, also here the elevation gain in the lower glacier part is comparably localized and largely due to the advance of the terminus.

*Fig. 10: 1D profile elevation changes compared to SRTM*

**South Chongtar** shows profiles (Fig. 10c) and surface change patterns (Fig. 11) that are in line with a typical surge, maybe apart from the fact that the thickening of the upper glacier regions is limited. The 2000 to 2010 change map (Fig. 11a) shows a slightly bluish upper part and some artefacts. Over the longer 2000 to 2015 period the elevation gain from 4 to 12 km is about 20-30 m (Fig. 10c), but further down a significant surface lowering (>50 m) can be observed between 13 and 18 km. This lowering is also visible in the 2D map of Fig. 11a, marking at its upper point the position where the active ice starts, i.e. where the surface lowering is compensated by the mass flux. The 2020 surge moved ice between 3 and 8 km towards its lower part between 10 and 16 km, causing a surface elevation decrease of 20-40 m in the reservoir zone and an increase of up to 130 m at km 14.



533 The ICESat-2 data points constrain the surface elevation evolution in time (Fig. 10c): The tongue

534 was still only slightly thicker at 12 km in March 2019, as surface lowering of the upper part (at 5.7

535 km) had not started in February 2020 and the terminus had not advanced by March 2020. Between

536 6 and 14 km we find a smooth linear increase of the elevation differences (Fig. 10c) - but the ICE-

537 Sat-2 data points at 9.5 km show a slight surface lowering between December 2019 and August

538 2020, indicating that the surge front passed this part of the glacier already before the end of August

539 2000. The 2015 to 2020 elevation change map (Fig. 11c) reveals that elevation changes mostly

540 occurred over this time period. Due to the opposite elevation change pattern before 2015, elevation

541 changes over the full period 2000-2020 are less pronounced. The constantly down-wasting Sarpo

542 Laggo Glacier in the valley floor shows an elevation loss of up to 100 m over this period.

544 *Fig. 11: 2D elevation change maps*

546 **5.4 Volume changes**

547 In Table 3 the results of the calculated volume changes are listed, differentiated for the gain and

548 loss part. They add some quantitative information over a larger part of the glacier surface (see Fig.

549 S5). With the timing of the DEMs not always synchronous with the start/end of a surge, the calcu-

550 lated values can be underestimated due to the overlap of surge phases. For example, the volume

551 gain in the lower part of South Chongtar from 2000 to 2020 includes the volume loss between

552 2000 and 2019. For this reason we only analyse the 2015 to 2020 changes for South Chongtar

554 For NN9 no mass transfer from an upper region is found. We have a near zero mass loss compared

555 to a clear volume gain of 0.03 km$^3$. For the continuous advance/surge of North Chongtar the vol-

556 ume gain is a bit higher than the loss resulting in a small overall volume gain over the full 20-year

557 period (Fig. 11d). However, Fig. 11c reveals that compensation effects are included. Between

558 2015 and 2020 some of the volume gain from the period before has already started thinning. The

559 volume gain part for South Chongtar is about 10 times higher compared to North Chongtar and

560 NN9. However, there is also considerable volume loss at higher elevations compensating about

561 half of the gain. To put just the volume gains of these five glaciers (+0.46 km$^3$) into perspective,

562 the (uncompensated) volume loss of Sarpo Laggo Glacier over the full period (-0.47 km$^3$) is the

563 same.


565 *Table 3: Volume changes*


567 **5.5 Sensor intercomparison**

568 **5.5.1 Velocities**

569 As can be seen in Fig. 7a, velocity values derived from the 15 m resolution Landsat 8 panchro-

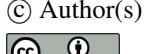



matic band for the period July 2018 to July 2019 are the same as from 10 m resolution Sentinel-2
data for the period August 2018 to August 2019. Both lines are basically on top of each other and
the average differences are insignificant. The same applies to TSX velocities from April to May
2014 compared to annual mean values from Landsat 8 over the July 2013 to July 2014 period, the
latter curve being on top of TSX in Fig. 7a (apart from the region between 8 and 9 km).

The Planet cubesat images cover only the lower part of the glacier. Here, the Planet velocity (Fig.
S9) reveals the same increase/decrease pattern as the Sentinel-2 velocity profile (Fig. 7b). Direct
comparison of flow velocities reveals only small differences (Fig. S10) that can be related to
slightly different time intervals. This is different when comparing Sentinel-1 derived velocities
with Sentinel-2 (Fig. S11). The large image template sizes of 128 x 64 (450 m x 900 m) for South
Chongtar (tongue width 800 m) result in a strong underestimation of Sentinel-1 velocities with
errors much greater than those indicated in previous studies for larger Arctic glaciers (Paul et al.,
2017; Strozzi et al., 2017). The density of the information is also very low compared to Sentinel-2,
indicating that Sentinel-1 data do not reveal sufficient detail about the surge.

**5.5.2 Elevation changes**
Of the seven analysed elevation datasets, ICESat-2 elevation profiles show most detail compared
to the DEMs and is also resolving small surface features such as crevasses and séracs (Fig. S6).
Both the weak and the strong laser beams of ICESat-2's three beam pairs provide equally good
data in the snow-covered accumulation areas (top panels in Fig. S6). On darker and more rugged
surfaces the weak beam yields considerably fewer photon returns than the strong beam (bottom
panels in Fig. S6).

Elevation differences between the HMA DEM and the SPOT DEM from 2015 are depicted in Fig.
S12. A small advance of North Chongtar and a slight elevation increase on South Chongtar within
the few months' time gap is visible. The latter is confirmed by the cross transects in the middle
panels in Fig. S6. In contrast, the elevations of the two 2015 DEMs agree very well for the tran-
sects in the upper accumulation area (top panels) and the down-wasting tongue in the main valley
(bottom panels). Apart from artefacts and local differences in very steep terrain, elevations of the
DEMs from 2015 agree very well both on and off glaciers.

The elevation changes derived from the ASTER DEM time series by Hugonnet et al. (2021)
shown in Fig. S13 are similar to the time series we analysed from SRTM, SPOT and the HMA
DEMs (Fig. 11). The ASTER DEMs have more artefacts and local differences, in particular in
very steep terrain. In contrast, the strong spatial filtering inherent in the ASTER dataset, smooth-
ens artefacts and data gaps off and to some degree also details on glaciers. Locally, the ASTER
data set is less complete, e.g., the advance of North Chongtar is not well covered and the advance



of the glacier NN8 is not visible.

There are no further insights when splitting the 2000-2010 period into a 2000-2004 and 2005-2009
period, but the 2000 to 2019 period from ASTER (Fig. S13f) reveals the up to 40 m elevation in-
crease in the upper region of South Chongtar. This 'reservoir zone' seemingly stretches over the
entire upper glacier rather than being an isolated region. In 2019 the surge has not started, so the
strong elevation loss in its lower part from post surge down-wasting is also very prominent. Eleva-
tion gain from 2000 to 2019 is also visible for the upper part of Sarpo Laggo Glacier and the lower
part of Moni Glacier.

**5.6 Uncertainty assessment**
**5.6.1 Glacier length changes**
Uncertainties of the length changes are estimated to be in the order of one image pixel, i.e. 60 m
for MSS, 30 m for TM, 15 m for OLI pan and 10 m for Sentinel-2. As frontal advances have only
been measured for a change of at least 3 to 4 pixels, the given values should be well outside the
uncertainty range in most cases. However, the calculated frontal advance rates for glacier NN9 and
North Chongtar (Fig. 4b) show fluctuations. These can be attributed to the measurement uncertain-
ties so that in reality the increase might have been smoother and more gradual. There is thus some
caution to not over-interpret the details of the change rates.

**5.6.2 Flow velocities**
The displacements measured by Landsat over the selected stable areas show median values close
to the expected value of 0 m d$^{-1}$, with a MAD between 0.01 and 0.04 m d$^{-1}$, as reported in Table S2.
Among the Landsat data, Landsat 7 shows the smallest standard deviation based on the MAD. For
Sentinel-2, the uncertainties of the displacement on stable terrain are lower for the pairs with a
time interval of approximately a year. For these pairs, the median and the MAD of the velocity are
of the same order of magnitude as the Landsat results. For shorter time intervals (5 to 45 days), the
Sentinel-2 velocity shows medians between 0.15 and 1.58 m d$^{-1}$ with a maximum MAD of 1.39 m
d$^{-1}$. Displacement from Planet data gives the largest error with medians and MAD values ranging
from 0.3 m d$^{-1}$ to 2.50 m d$^{-1}$. One pair showed a significantly higher error with a median value of
8.64 m d$^{-1}$ and a corresponding MAD value of 4.76 m d$^{-1}$, which is in a similar order of magnitude
to the displacement measured in the centre line of the glacier (13.89 m d$^{-1}$). TSX revealed the low-
est uncertainty with values of both median and MAD close to 0 m d$^{-1}$.

**5.6.3 Elevation data**
The median elevation differences to the reference DEM (HMA DEM) are 1.02 m (SRTM), 1.03 m
(SPOT 2010), -0.12 m (SPOT 2015) and 1.08 m (SPOT 2020), with standard deviations of 3-15 m
(Table S3). Also mean elevation differences, which are more sensitive to extreme values, are <1.4



m for all DEM difference pairs except for the SPOT 2020-SRTM2000 DEM pair (2.4 ±8.8 m).
These are small differences and fully within the range of expected uncertainties (after successful
co-registration), considering the very steep and rugged terrain. We found no indication of remain-
ing horizontal shifts between the DEMs (this would be visible as an aspect-dependent pattern in
Fig. 11). The comparison of the SPOT 2015 and HMA 2015 DEM (Fig. S12) shows a minor tiling
effect caused by the composite nature of the HMA DEM in the upper accumulation areas of North
and South Chongtar. The mean uncertainty of the ATL06 ICESat-2 data was ±5.37 m. However,
we assume that ATL03 elevation uncertainties are in the order of decimetres on the relatively
smooth glacier surface.


## 6.  Discussion


### 6.1 Interpretation of the surges

The contrasting surge behaviour of North and South Chongtar glacier is remarkable as the two
glaciers with the likely highest (South Chongtar) and lowest (North Chongtar) flow velocities and
advance rates during a surge (in the entire Karakoram) can be found side-by-side. At first glance it
seems that the sudden and short-lived surge of South Chongtar is hydrologically controlled (Alas-
ka type), whereas the neighbouring North Chongtar surge seems thermally controlled (Svalbard
type). However, as Quincey et al. (2015) noted, this simplified picture does not hold for many
glacier surges in the Karakoram, which often show characteristics of both types. For example, the
South Chongtar surge reached its maximum flow velocities in summer rather than winter and their
drop is very slow rather than fast. Moreover, flow velocities increased slowly, constantly and over
large parts of the glacier rather than being located at a clearly localized surge front. These observa-
tions better fit to a thermally controlled surge and implies that both mechanisms apply and the
surge mechanism could be named 'hybrid'.

The slow and near constant advance of North Chongtar and NN9 might not even be classified as a
surge, but given that both glaciers also developed nearly all characteristics of a surge at some
point, the former advance phase might be seen as a part of the surge. Still, from the evolution of
advance rates or flow velocities alone it is nearly impossible to pin down the exact surge onset for
North Chongtar. Morphological changes (heavy crevassing, shear margins) indicate that this might
have happened around 2010, but considering the near linear increase of advance rates after 1996,
one might assign the onset also to that year. In any case, the more or less constant advance for
more than 30 years before 1996 is exceptional and only comparable to the very slow advance of
Maedan Glacier in the neighbouring Panmah region that also started in the 1960s, before advance
rates considerably increased in the mid-1990s and the glacier started surging (Bhambri et al. 2017;
Paul, 2020). Such prolonged advances might also be a consequence of a positive mass balance that





one glacier converted to a continuous advance and another one to a surge (Lv et al., 2020). At least
the elevation change pattern of North Chongtar over the 2000-2020 period reveals a clear and typi-
cal redistribution of mass from a higher reservoir zone to a lower receiving zone.

This is different for NN9, which only shows elevation increase in its lower part over this period
without any measurable surface lowering higher up. This rather unique criterion for surge identifi-
cation fails here and would exclude the glacier from being surge type. However, a different im-
pression emerges when looking at the temporal evolution of advance rates and flow velocities. In
2016 the former increased considerably from about 40 m y$^{-1}$ to more than 1 km y$^{-1}$ and the mor-
phology of the surface changed from rather smooth to highly crevassed. Measurable flow veloci-
ties increased in 2019 from 0.2 to 0.8 m y$^{-1}$ and the Landsat 8 image pair from 2018 to 2019 (Fig.
5d) also reveals an increase. With its recent rapid advance, the glacier has now reached its former
1971 maximum extent and also looks the same in terms of a completely crevassed surface (Paul,
2020). The slow advance might have resulted from a positive mass balance but could also be a
thermally controlled surge. However, the recent increase in advance rates could also be due to a
hydrologically controlled surge and/or due to the steep slope and dynamic effects. Compared to
North Chongtar, the switch from advance to surge occurred much more sudden.

For South Chongtar the situation is clearer as its rapid advance and more than 100-fold increase in
flow velocities (from 0.2 to more than 25 m/d) is typical for a hydrologically controlled surge with
increasing basal water pressure. We assume that its thin lowest part was frozen to the bed (e.g.
Obu et al., 2019), effectively blocking water release for some time. The interesting points of the
current surge are: (a) the gradual increase of flow velocities in the region above its fixed terminus
(at 13.5 km), (b) the extreme velocity increase from July to September 2020, (c) the high maxi-
mum velocities of 30 m d$^{-1}$, (d) the location of the maximum away from the centre, and (e) the
more or less constantly high flow velocities over large parts of its length from January to May
2021. The latter is responsible for the ongoing mass transport and advance of the terminus and im-
plies that basically the entire glacier was activated by the surge. As mentioned above, points (a)
and (e) are more typical for thermally controlled surges so with both characteristics this surge can
be classified as hybrid. That velocities increase from the centre to the boundary of a glacier (Fig.
9) is likely rather unique. We assume this is caused by the surrounding topography, i.e. the change
of flow direction from northwest to north imposed by the mountain walls. The centre of the ad-
vancing terminus collided with the southern rock wall and was then diverted to a different direc-
tion. As the glacier was likely sliding over its full width, the resistance at the boundaries was likely
limited.

Maximum surface flow velocities of 30 m d$^{-1}$ are only visible with Planet and to the edge of the
glacier (Fig. 9), Sentinel-2 values peak at 27 m d$^{-1}$. This is likely due to the higher resolution of





Planet compared to Sentinel-2 and hence to the smaller region used for spatial averaging. Also the
shorter time period considered (3 days) might play a role. Whereas high flow velocities of about
15 m d$^{-1}$ have been reported previously (Quincey et al., 2015; Paul et al., 2017; Bhambri et al.,
2020), values above 25 m d$^{-1}$ are only rarely observed in the Karakoram (Rashid et al., 2020). The
latter study reports values near 50 m d$^{-1}$ for the last surge of Shispar Glacier (derived from 3 m
Planet data), but the flow fields look a bit 'bumpy' and image processing artefacts might have con-
tributed to the high values. We assume that the rapid increase in flow velocities during Ju-
ly/August was due to additional lubrication from summer surface melt water.

In principle, a surge simply moves mass downstream implying that the net volume change should
be about zero. However, if surges take place over very long periods (>5 years) there will also be a
signal from the usual ablation and accumulation. Moreover, for DEMs derived from optical sen-
sors problems in snow covered or steep terrain (shadow) exists that might create data gaps in the
region where the mass has been removed (or where mass gain took place before a surge). Both
effects can create biases leading to over- or underestimation of calculated volume changes. These
apply also to the changes calculated over ten-year periods and the SPOT DEM from 2010 that had
data voids in the steep upper regions of some glaciers. In consequence, volume changes calculated
with this DEM are incomplete and need to be interpreted with care. However, as both positive and
negative changes take place in regions of increased uncertainty, the net effect is likely small.

**6.2 Sensor capabilities and limits**
The sensor intercomparison revealed a very good agreement between the velocity data derived
from TSX StripMap mode and Sentinel-2 with Landsat 8 (Fig. 7a), as well as between Sentinel-2
and Planet (Fig. S10). This confirms that all three optical sensors can be used to derive the tem-
poral evolution of flow velocities – cloud cover, snow conditions and cast shadow permitting. The
key point is the choice of the temporal baseline of image pairs as a function of glacier surface
changes, sensor resolution and the targeted velocity field. At 20 m d$^{-1}$ a 5 (3) day interval is equiv-
alent to a change by 10 pixels with Sentinel-2 (20 with Planet in 3 days). At 0.1 m d$^{-1}$ the dis-
placement is about 35 m (3 Sentinel-2 pixels) after a year, which is at the lower end of what is de-
tectable with offset-tracking.

Unfortunately, Sentinel-1 performed poorly on South Chongtar, mostly due to the fact that it is a
narrow tongue (width less than 800 m) situated between steep mountain flanks. Because of the
relative large size needed for the matching window (Fig. S4), too many non-moving off-glacier
pixels are included affecting the velocity retrieval considerably (Fig. S11). Also, the large and fast
surface changes on the rapidly surging glacier might have changed the backscatter patterns too
much to be tracked over time (Strozzi et al., 2017). The minimum width of a glacier to be reliable
monitored with Sentinel-1 in the Himalayas is likely around 2 km. On the contrary, TSX yielded



dense and consistent velocity values for all three glaciers (pre-surge-phase). As it seems, the map
in Fig. 5b captures nicely the flow acceleration of North Chongtar in 2012, which decreased after-
wards (Fig. 6b). The much noisier values from Landsat 8 in this figure (compared to Sentinel-2
and TSX) revealed that the 15 m resolution of the Landsat panchromatic band is seemingly insuf-
ficient to track displacements precisely. Note, though, that these comparisons are not strictly as the
sensors have different resolutions, and the datasets cover different phases of the surges and thus
different surface conditions.
The compared DEMs are of similar quality over glaciers, but the SPOT 2010 DEM used by
Gardelle et al. (2013) suffered from strong artefacts at steep slopes. The elevation values of the
SPOT 2015 and HMA DEM (which is also from 2015 in this region) are basically identical apart
from individual raster cells. So elevation changes from 2000 (SRTM) to 2015 (HMA DEM) can
also be derived from freely available DEMs. The SPOT 2020 DEM is of superb quality but the
raw image pair had to be purchased. For a study looking at specific glaciers this is certainly
worthwhile, but does typically hamper larger regions to be covered.
The surface elevation detail and accuracy of the freely available ICESat-2 ATL03 photon data sur-
passes all other datasets, including the SPOT 2020 DEM (Fig. S6). When combined with one or
several DEMs, the higher temporal resolution provides additional information on how the eleva-
tion changed in-between DEM time stamps. This may be very useful for slower changes or to fur-
ther constrain the onset/end of a rapid change, such as a surge. However, ICESat-2 only provides
elevation profiles with varying locations, which makes this data type more demanding to analyse.
The footprints of the ICESat-2 ATL06 time series alone are too sparse to derive any useful trends
in glacier surface elevation.
The DEM time series from ASTER images (Fig. S13) derived by Hugonnet et al. (2021) shows the
same trends as from the DEMs used here. They provide further information over the 2000–2005
and 2005–2010 periods, but miss the surge of South Chongtar as they end in 2019. On the other
hand, they cover a much larger area and clearly reveal the increase in surface elevation of South
Chongtar over the full 2000-2019 period. The coverage of the smaller glaciers is noisier with AS-
TER than with the DEMs we have used and locally values are missing, but the temporal evolution
over several larger glaciers can be well followed. Deriving further DEMs from future ASTER ste-
reo scenes might thus help to determine total volume changes after all surges have come to an end,
including the yet not visible volume loss in the reservoir zone of NN9.

**6.3 Uncertainties**

The one-pixel uncertainty in deriving terminus positions and length changes translates into an un-
certainty of the calculated advance rates. How large the uncertainties are, depends on the sensor



resolution and the time period between two measurements. It is assumed that at least a part of the
variation in the advance rates of NN9 and North Chongtar are due to these uncertainties rather than
real variability.

With the exception of the Planet data, the uncertainty of the velocity measured over stable terrain
by all sensors is one or two orders of magnitude smaller than the maximum displacement observed
on the glacier along the centreline, even for the two small glaciers NN9 and North Chongtar. For
them, cloud cover has been identified as a major challenge for optical sensors. In fact, the selection
of the satellite pair prioritized the reduction of cloud cover on South Chongtar rather than NN9 and
North Chongtar, which were rarely cloud-free. Hence, it is not only spatial resolution that is re-
sponsible for data limitations.

In general, the uncertainties of glacier flow velocity measurements are mainly related to co-
registration accuracy, orthorectification, the time interval between image pairs, surface conditions
(shadow, snow, etc.) and the spatial resolution of the images. The larger the time window between
two pairs, the smaller the uncertainty of the measured velocity. Despite the higher resolution, the
uncertainty is higher for Planet than for Sentinel-2. For Sentinel-2, the orthorectification error is
minimized because the imagery comes from the same relative orbit (Kääb et al., 2016). On the
contrary, we have different orbital paths between Planet image pairs and therefore further geomet-
ric corrections may be needed to minimize this error, as also suggested by Kääb et al. (2017) and
Millan et al. (2019). Also the very small stable terrain uncertainties of TSX are likely due to the
accurate co-registration of the image pairs.

The observed elevation changes exceed the DEM elevation uncertainties by an order of magnitude
or more, which makes our elevation change analyses very robust. For volume change studies, data
gaps in the DEMs and remaining blunders/bias from clouds or other sources cause greater uncer-
tainties than the elevation uncertainties themselves (McNabb et al. 2019). Data gaps occur, how-
ever, mostly in the accumulation areas due to reduced contrast over snow, more persistent cloud
cover and steeper terrain. Moreover, surface elevation tends to change much less here than it is the
case for the tongues, and uncertainties might become as large as the changes. The elevation accu-
racy of the ICESat-2 ATL03 product is clearly superior to all DEMs analysed within this study.


## 7. Conclusions



We have identified and presented an analysis of three glacier surges in the central Karakoram, all
taking place in the same region but with very different characteristics and possibly forcing mecha-
nisms. South Chongtar showed advance rates of more than 10 km y$^{-1}$, velocities up to 30 m d$^{-1}$ and



surface elevations rose by 200 m. The three times smaller North Chongtar has a slow and almost-
linear increase of advance rates (up to 500 m y$^{-1}$), flow velocities below 1 m d$^{-1}$ and elevation in-
creases of up to 100 m. The even smaller glacier NN9 changed from a slow advance to a full surge
within a year, reaching advance rates higher than 1 km y$^{-1}$, but showing the typical surface lower-
ing higher up only recently. Total length changes reached between 2 and 2.7 km for the three glac-
iers and the size of NN9 changed by more than 20%. For South Chongtar, maximum flow veloci-
ties are found near its southern boundary rather than in the centre.

At first glance, the surge of South Chongtar clearly resembles the classical Alaska type surge (hy-
drologically controlled), whereas North Chongtar and NN9 better fit to the Svalbard type (thermal-
ly controlled). However, the summer onset and slow velocity decay of the South Chongtar surge
and the sudden change in frontal advance rates of NN9 hint to the respective other type, resulting
in a change of characteristics. North Chongtar has not changed type but surge onset is difficult to
determine as advance rates increased linearly, morphological changes developed slowly and a 50-
year advance might also be called a surge. When the definition of a surge is stricter, we would as-
sign the surge onset of NN9, North and South Chongtar to 2017, 2005-2010 and August 2019, re-
spectively. We speculate that the thin, lower part of South Chongtar was cold ice frozen to the bed,
reducing possibilities for the terminus to advance and causing basal pressure to strongly increase.

The sensor intercomparison revealed that Landsat 8 and Sentinel-2 are difficult to be used jointly
for determination of geometric changes as their geolocation differs (>30 m). Flow velocities
agreed well across sensors for South Chongtar, except for Sentinel-1 that had problems due to its
narrow tongue (800 m). However, the backscatter intensity images provided a time-series of surge
evolution at a near constant interval that is undisturbed by clouds. At the two smaller glaciers NN9
and North Chongtar, the optical sensors still provided reasonable and consistent flow velocities,
but limits due to spatial resolution and cloud cover became visible (more noise). The TerraSAR-X
acquisitions in StripMap mode revealed by far the best results and depicted the surge of North
Chongtar accurately.

After proper co-registration, all DEMs provided useful results to track elevation and volume
changes, independent of glacier size. The two SPOT DEMs from 2010 and 2015 suffered from
artefacts at steep slopes, but the latter compared very well to the HMA-DEM. The high-resolution
SPOT6 DEM from Oct 2020 had impressive quality and allowed an accurate calculation of the
volume change of all glaciers up to this point in time. The very precise ICESat-2 elevation profiles
provided additional information in space (glacier surface details) and time (between the DEMs)
that matched well to the other datasets. The ASTER DEM time series missed detecting local
changes of smaller glaciers, but provided a larger overview and complementary information on
cumulative elevation changes shortly before the surge of South Chongtar started.



All three glaciers are still advancing and South Chongtar and NN9 are now colliding. The bulldoz-
ing of the South Chongtar terminus into the down-wasting ice of Sarpo Laggo Glacier is already
creating interesting morphological changes. North Chongtar might again reach the floor of the
main valley as in the 1930s, but this could take some more years. We conclude that the past and
further evolution of these and other glacier surges can be well observed from satellite data, at best
by combing all available datasets.

**Supplement**

The supplement related to this article is available on-line at: TBD

**Author contributions**

F.P. detected the surges, lead the writing and analysed changes in extent and morphology. L.P.
contributed equally, derived the optical velocity data and prepared all related tables and figures;
D.T. derived and combined the elevation change data. All authors contributed to the writing, dis-
cussion and editing of the text.

**Code and data availability**

Data processing has been performed using freely available (e.g. CIAS, MicMac, geopan-
das/rasterio/shapely) or in-house software (for SAR offset-tracking). Also most of the datasets
used here are freely available (e.g. Landsat, Sentinel-1/-2, Planet, ICESat-2, SRTM and HMA
DEMs, glacier outlines), except TerraSAR-X data (ordered from DLR) and the SPOT2020 DEM
(ordered form Airbus). The SPOT DEMs from 2010 and 2015 were provided by E. Berthier.

**Competing interests**

The authors declare that they have no conflict of interest.

**Acknowledgements**

This study has been performed in the framework of the ESA project Glaciers_cci
(4000127593/19/I-NB). We thank Etienne Berthier for providing the SPOT 2010 and 2015 DEMs.
We also acknowledge free access to Sentinel-1 and -2 data from Copernicus, Landsat from USGS,
Planet from Planet, the SRTM DEM from USGS, the HMA DEM from NSIDC, and glacier out-
lines from GLIMS. This study would not have been possible otherwise.

**Financial support**

This study has been supported by the ESA project Glaciers_cci (grant no. 4000127593/19/I-NB).



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



**Tables**

*Table 1: Characteristics of the three investigated glaciers using outlines modified from the*
*GAMDAM2 glacier inventory (Sakai et al. 2019) and digitized in this study. Elevations refer to the*
*SRTM DEM. Values given for 'min/max' refer to the minimum and maximum extent of a glacier*
*shortly before and after a surge, respectively.*

| | NN9 | North Chongtar | South Chongtar |
|---|---|---|---|
| Size (min / max) | 3.93 / 4.78 km$^2$ | 9.16 / 10.15 km$^2$ | 31.09 / 34.23 km$^2$ |
| Size change (km$^2$ / percent) | +0.85 km$^2$ / +21.6% | +0.99 km$^2$ / +10.9% | +3.14 km$^2$ / +10.0% |
| Elevation (highest / mean) | 6450 / 5620 m | 6810 / 5860 m | 7230 / 5920 m |
| Lowest elevation (min / max) | 4430 / 5075 m | 4440 / 5015 m | 4545 / 4400 m |
| Length (min / max) | 3.25 / 5.5 km | 4.75 / 6.8 km | 14.4 / 17.1 km |
| Changes (min. elev. / length) | 645 / 2250 m | 575 / 2050 m | 145 / 2700 m |
| Slope / Aspect | 31.5 / SE | 28.8 / NW | 25.1 / NW |
| Previous surge | 1961-1971 | 1920s | 1966-68 |
| Surge repeat cycle | 40-50 y | 90 y? | 54 y |
| This surge | 2000-today | 1965-today | 2020-today |
| Characteristics | Compact dual-basin valley glacier with prominent medial moraine | Dual-basin valley glacier with one major tributary forming a prominent medial moraine | Long and flat single-basin valley glacier with three tributaries (one resulting in a short medial moraine) |



*Table 2: Overview of the DEMs used to determine elevation changes of the glaciers in the study*
*region and the additional ICESat-2 dataset.*

| Nr. | Name (short) | Type | Reso-lution | Date | Source | Comments |
|---|---|---|---|---|---|---|
| 1 | SRTM 1 | SAR | 30 m | Feb. 2000 | USGS, doi: 10.5066/f7pr7tft | C-band w/ penetration |
| 2 | HMA-DEM | OPT | 8 m | Feb.-Aug. 2015 | NSIDC, doi: 10.5067/KXOVQ9L172S2 | 7 months composite |
| 3 | SPOT 2010 | OPT | 30 m | 31 Oct 2010 | Gardelle et al. 20013 | SPOT 5 HRS |
| 4 | SPOT 2015 | OPT | 30 m | 13 Oct 2015 | Berthier & Brun 2019 | SPOT 6 |
| 5 | SPOT 2020 | OPT | 10 m | 20 Oct 2020 | Ordered from Airbus | SPOT 6 |
| 6 | ASTER | OPT | 30 m | 2000-2019 | Hugonnet et al. 2021 | 5y elevation changes |
| 7 | ICESat-2 | LIDAR | 0.7 m | 3.12. 2018 – 5.11.2020 | NSIDC, nsidc.org/data/icesat-2/data-sets | Version 4, 14 tracks, over glaciers only |



*Table 3: Calculated volume changes (in km$^3$) for six glaciers and different periods as obtained*
*from the respective DEMs. Grey umbers in italics denote results that might be impacted by*
*artefacts.*

| Nr. | Glacier | Period | Gain | Loss | Total |
|---|---|---|---|---|---|
| 1 | South Chongtar | 2015-2020 | 0.3444 | -0.1760 | 0.1684 |
| 2 | North Chongtar | 2000-2020 | 0.0466 | -0.0365 | 0.0102 |
| 3 | NN9 | 2000-2020 | 0.0356 | *-0.0017* | 0.0339 |
| 4 | NN8 | 2000-2010 | 0.0175 | -0.0167 | 0.0008 |
| 5 | NN7 | 2000-2010 | 0.0146 | -0.0304 | -0.0158 |
| 6 | Sarpo Laggo | 2000-2020 | *0.0024* | -0.4708 | -0.4684 |







**Figures**

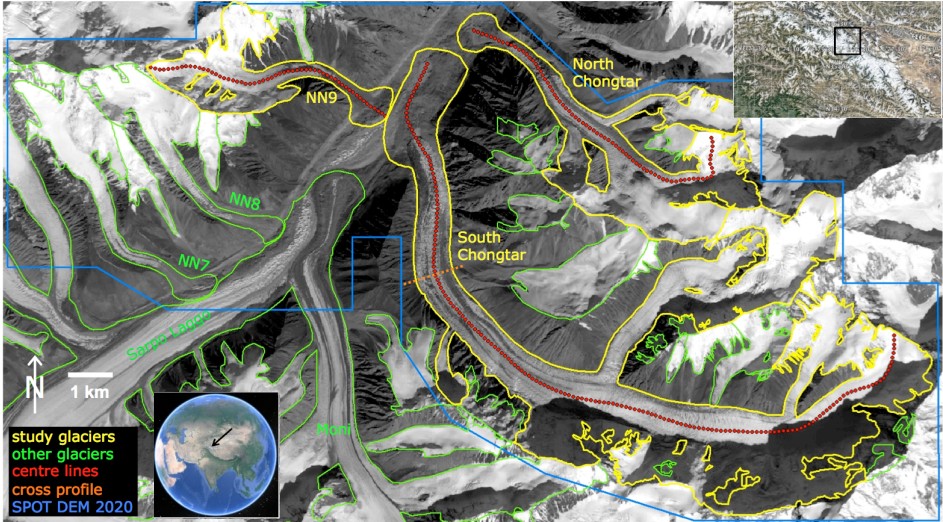

*Fig. 1: Overview of the study region showing the location of the Karakoram Mountains (inset, lower left) and of the study region (inset upper right), outlines of the investigated glaciers (yellow), other glaciers (green), centre lines (red), cross-profile line (orange) and extent of the SPOT 2020 DEM perimeter (blue). The satellite image in the background is the panchromatic band from Landsat 8 acquired on 21 Oct 2020 (Landsat data: earthexplorer.usgs.gov), the two insets are screenshots from © Google Earth.*

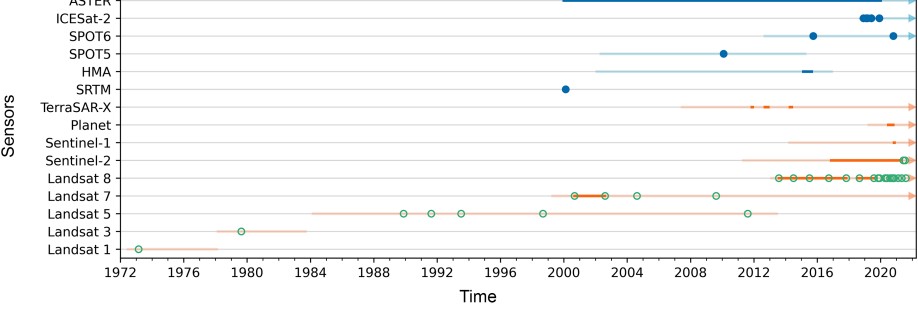

*Figure 2. Timeline of the temporal coverage of the satellite sensors used (light line) and dates and time series selected for the analysis (lines or dots). Lines and dots in dark blue indicate the elevation change analysis, orange lines the velocity analysis, and green dots the glacier extents.*






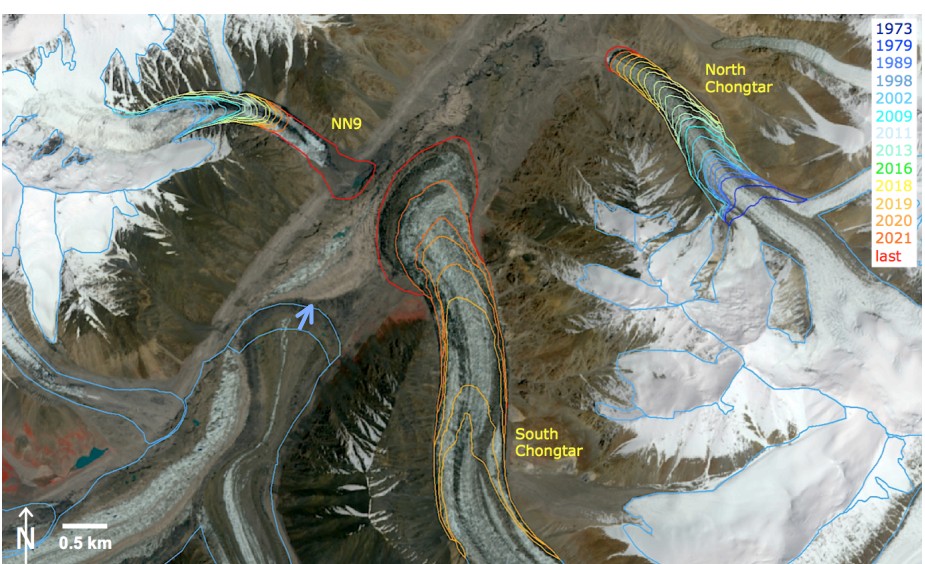


*Fig. 3: Temporal evolution (colour-coded dates) of glacier extent for the three glaciers (NN9,*
*North Chongtar, South Chongtar) investigated here. For comparison, the displacement of the*
*terminal lobe of Moni Glacier from 2000 to 2020 is also shown (arrow). 'Last' is referring to 30*
*September 2021. Background: Sentinel-2 image acquired on 16 July 2021 with bands 8/4/3 as*
*RGB (Copernicus Sentinel data 2021).*


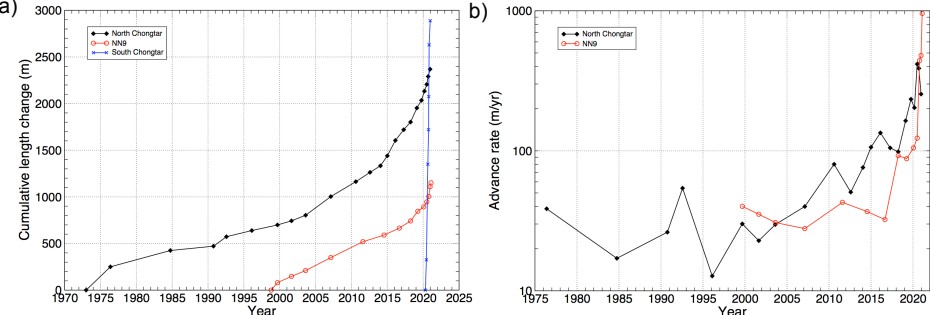


*Fig. 4: Terminus changes for the investigated glaciers. a) Cumulative length changes (the retreat*
*phase of South Chongtar before 2020 is not shown), b) advance rates.*



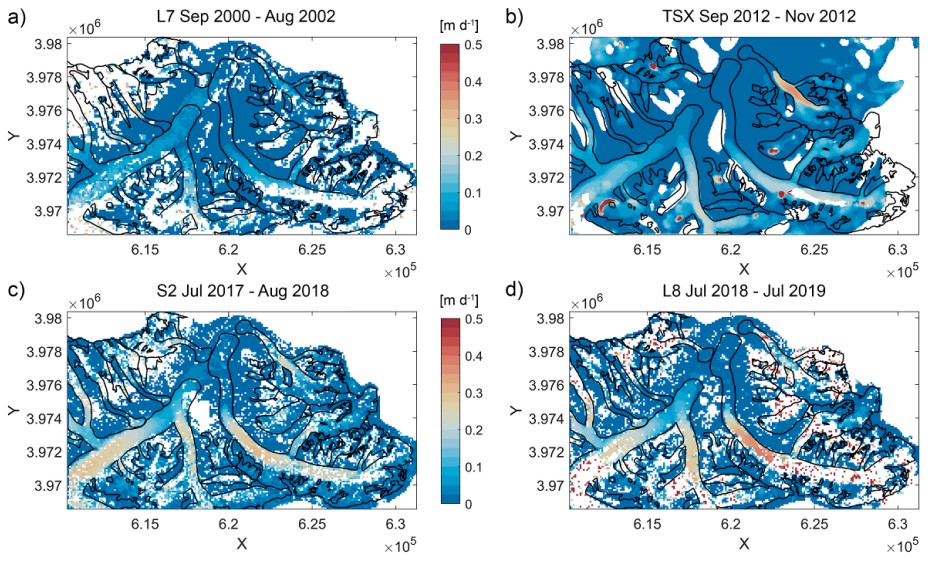


*Fig. 5: Temporal evolution of 2D surface flow velocities for the three glaciers before 2020 derived*
*from a) Landsat 7, b) TerraSAR-X, c) Sentinel-2, and d) Landsat 8. The dates of the compared*
*images are given at the top of each panel.*

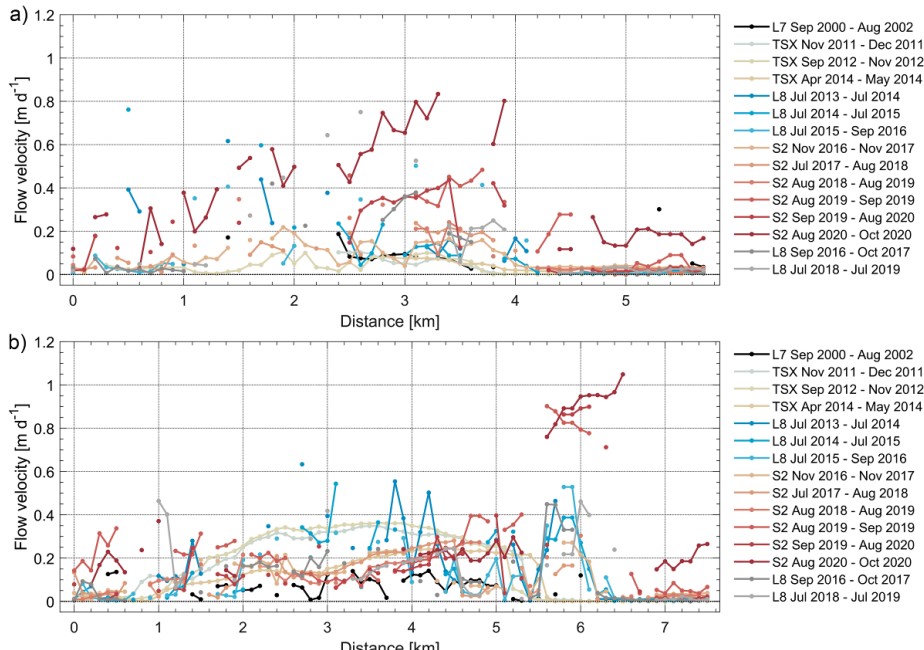


*Fig. 6: Temporal evolution of 1D flow velocities along a centre-line starting at the highest point of*
*each glacier for a) glacier NN9 and b) North Chongtar. Satellite names: L7/L8: Landsat 7/8, TSX:*
*Terra-SAR-X, S2: Sentinel-2.*

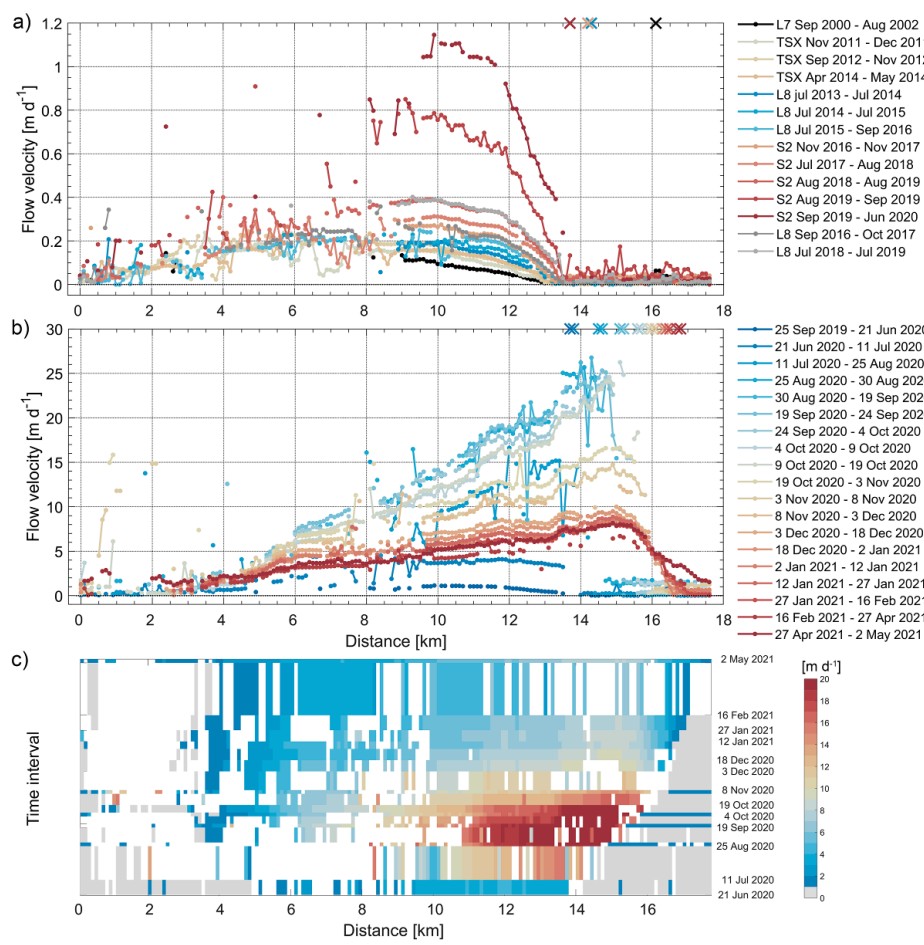


*Fig. 7: Temporal evolution of flow velocities for South Chongtar Glacier from its highest point to*
*its terminus, its location is indicated by an 'x' at the top of panels a) and b). a) Pre-surge values*
*along the centre-line as derived from different satellites (names see Fig. 6). b) As a) but during the*
*surge and derived from Sentinel-2 only, c) Hovmöller diagram of the surge phase. In this plot grey*
*values are below 1 m d⁻¹, white indicates no data.*







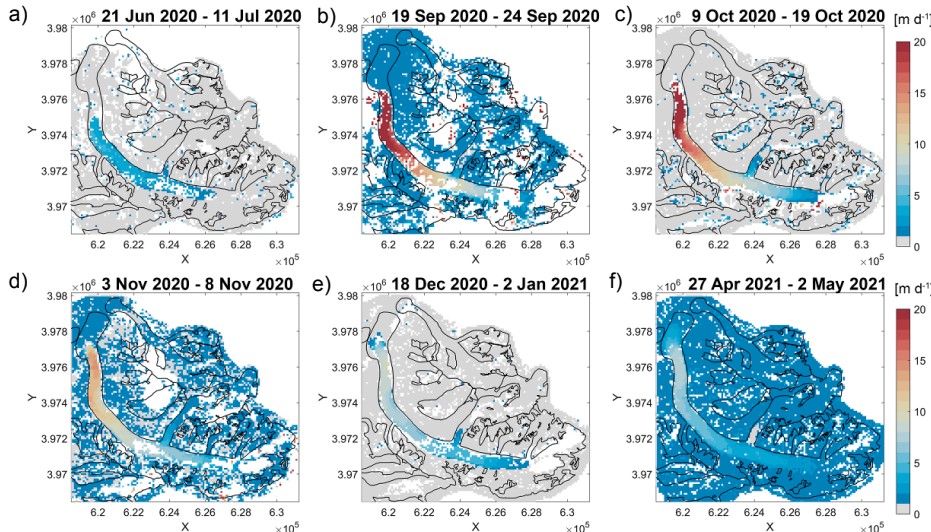


*Fig. 8: Temporal evolution of 2D flow velocities for South Chongtar Glacier during its surge as*
*derived from Sentinel-2. The dates of the respective Sentinel-2 pairs are given at the top of each*
*panel.*


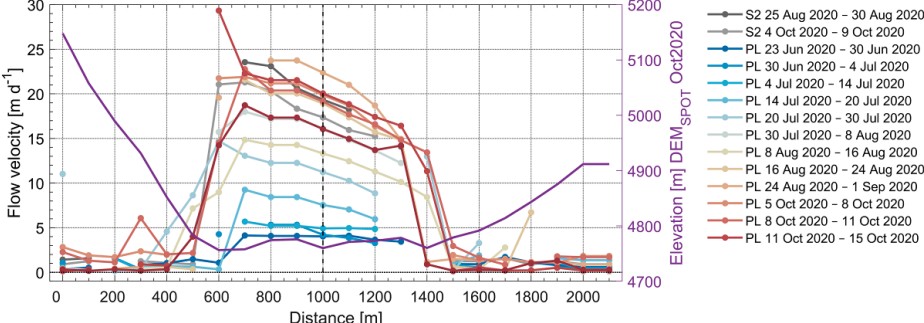


*Fig. 9: Cross-profile surface flow velocities for South Chongtar Glacier derived from Planet and*
*comparison with Sentinel-2. The vertical dash line indicates the location of the centerline.*





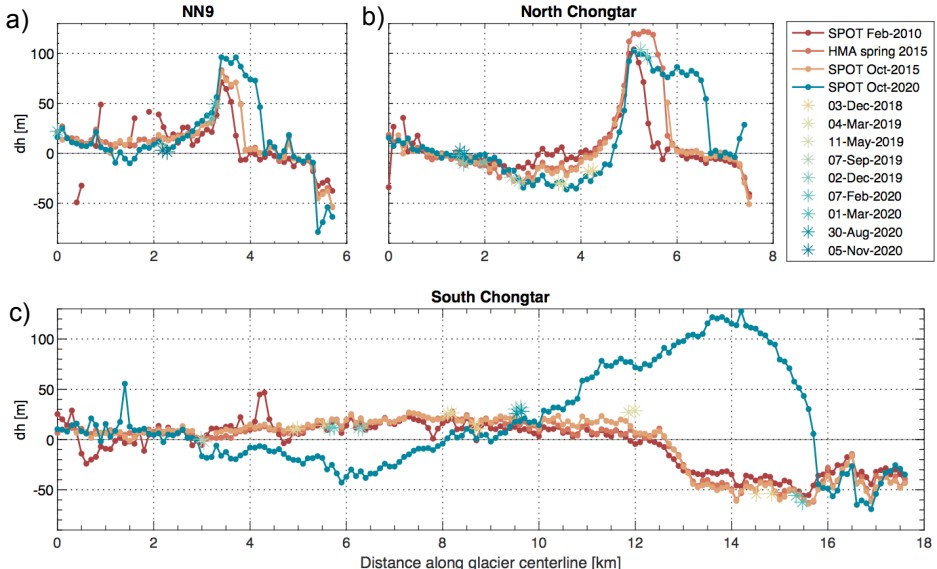


*Fig. 10: Elevation differences along the glacier centerlines in respect to the SRTM DEM from*
*2000 for the three investigated glaciers, namely a) NN9, b) North Chongtar, and c) South*
*Chongtar glaciers. The star (\*) markers and dates in the legend correspond to ICESat-2 elevation*
*differences in respect to the SRTM DEM. Note that due to the different track locations, only some*
*of the dates shown in the legend are present in each panel.*

1221

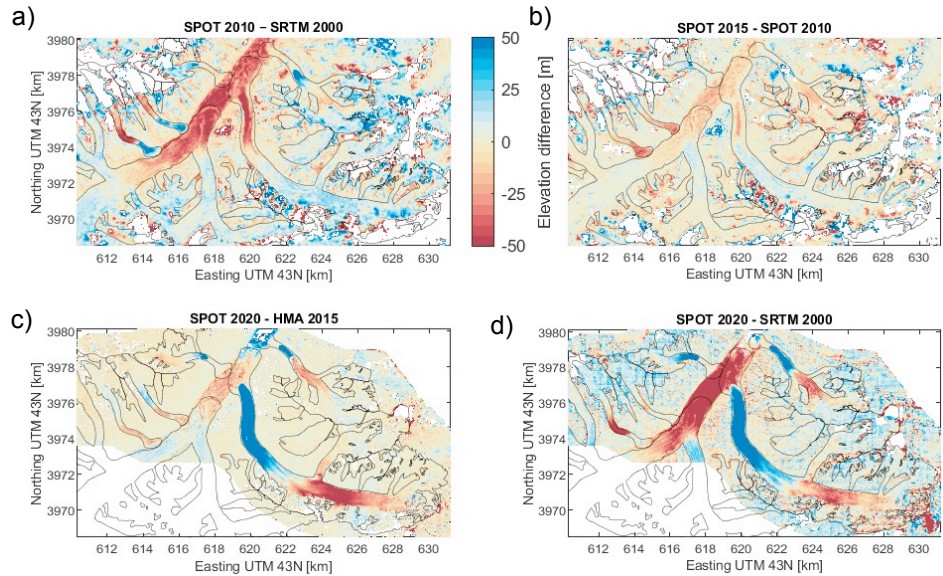

1222

*Fig. 11: 2D elevation difference maps over the study region. a) SPOT 2010 – SRTM 2000, b)*
*SPOT 2015 – SPOT 2010, c) SPOT 2020 – HMA 2015, d) SPOT 2020 – SRTM 2000. A*
*comparison between the SPOT 2015 and the HMA DEM from 2015 is shown in Fig. S13.*