# Peer review of "Three different glacier surges at a spot"

_The Cryosphere, 2021_

## Author Comment (AC1)

**Response to the reviewers comments**

**RC1: 'Comment on tc-2021-370', Anonymous Referee #1**

This paper provides an interesting review of the contrasting behaviour of adjacent surge-type glaciers in the Karakoram, and in particular how a very slowly surging glacier (likely controlled by changes in basal thermal conditions) can exist next to rapidly advancing ones (likely controlled by changes in basal hydrological conditions). These changes are well documented and illustrated through changes in terminus position, surface elevation and ice velocity, both in the main paper and the extensive supplemental material. I haven't seen many comparisons of surging glaciers which contrast in their behaviour over such a short distance before, so this provides the primary novel contribution of this study. However, the current title doesn't well describe this: from the title I expected the paper to provide a methodological assessment of the pros and cons of different remote sensing methods for detecting glacier surges (and how some methods might 'miss' features due to low spatial or temporal resolution), but the paper doesn't really do this; the main finding is that most methods actually work pretty well for detecting surges and their associated velocity and elevation changes. So I recommend that the title be changed to something that better describes the actual content of the paper, which if we rearrange the wording from the first sentence of the Conclusions (L833-835) could be something like: 'Contrasting characteristics and forcing mechanisms of three glacier surges in the central Karakoram'. Rearrangement of a few sentences or sections in the main text to better reflect this focus could help.

*--- We thank you for the detailed and very constructive review! We had indeed some discussions about the title and decided to be more catchy than descriptive. Although we did not mention the study region, we had the impression that the most important aspects are covered: We present satellite-based observations of three glacier surges taking place in the same region and show what the different sensors are able to observe and where they start to struggle. These are the two major goals of the study. We agree that by writing what satellites 'detect' instead of 'observe' our expectations regarding the contents would have been the same as yours. The interpretation of the observations (e.g. the forcing mechanisms part) is a more speculative part of the study that we would thus prefer not to claim already in the title.*

My specific comments are below. The biggest one is that the Discussion and Conclusions would benefit from better referencing to previous work.

*--- We have added further references to the discussion section but prefer to not do this for the conclusions, as in our opinion the conclusion should not introduce new or selectively repeat references. We acknowledge that this might be a personal view.*

**Detailed comments**

L27/28: provide some more information about the 'full suite of satellite sensors' and 'DEMs from different sources' that you used – e.g., names of sensors, resolution, repeat coverage period. This is one of the most useful things that I would like to know when reading the abstract, more so than some of the other background information currently provided.

*--- Done.*

L43: fit neatly with what? Clarify

*--- 'to the other DEMs' (added)*

L52: change 'provides' to 'provide'

*--- Done.*

L72-75: it might be useful to add reference to Hewitt (2005) here, which provides some discussion of the different potential causes of Karakoram surges, particularly in relation to changes in basal thermal regime: Hewitt, K. 2005. The Karakoram Anomaly? Glacier Expansion and the 'Elevation Effect,' Karakoram Himalaya. Mountain Research and Development, 25(4), 332-340.

L107: I think that you mean 35.94N, not 5.94N!
*--- Yes, indeed! Thank you for spotting.*

L163: seems to be missing 'with' before 'the Landsat 7…'
*--- Yes, corrected.*

L173: I don't really understand what 'virtual' refers to here. Either needs more explanation, or stating something like 'we digitized likely maximum extents' would seem to be clearer.
*--- Agreed, changed to your suggestion.*

L177: What does 'spatially consistent' refer to? E.g., orthorectified?
*--- No, both datasets (Landsat and Sentinel-2) are orthorectified but there was a spatial shift between Landsat and Sentinel-2 for the USGS Collection 1 scenes we used here. There is no shift when only using Landsat or Sentinel-2, i.e. for the same sensor the scenes are spatially consistent. We have rewritten this part to be clearer.*

L168-201: For readers who may be unfamiliar with the satellite sensors used, please make it clear as to which are optical and which are SAR. Also provide the resolution for the optical scenes (you currently only mention this for SAR).
*--- Agreed and added.*

L188: please state whether you selected different image types from particular times of the year. E.g., was SAR data preferentially acquired in the winter? Was optical data preferentially collected under summer snow-free conditions?
*--- We have added information about the acquisition season in the main text. Details can be found in Table S2. For the surge phase of South Chongtar we could use also winter images from optical sensors as snow cover was limited. More difficult is the monsoon season (due to clouds).*

L207: the SPOT5 DEM is stated as being from January 2010 here, but the date is listed as 31 Oct 2010 in Table 2
*--- Thank you for spotting this mismatch, it has been corrected now (October was correct).*

L216: Hugonnet et al. (2021) list their datasets as being from 2000-2004 and 2005-2009
*--- Yes, indeed. When downloading the data one gets a 1.1.2000-1.1.2005 period for the 2000-2004 dataset, likely to indicate that values have been extracted for full year periods rather than the usual autumn to autumn periods. However, to avoid confusion we have changed it now here and in Fig. S13 and mentioned the different reference periods.*

L244: state why a two-pixel change is important. Related to the above comment, the resolution of the image sources is also needed to understand what two pixel relates to in real world distance for different image types.
*--- Yes, agreed. This matters in particular for the slow advance of North Chongtar where a 120 or 60 m advance will take some time. We have added this information and explained the two pixel threshold.*

L260: when you mention the offset-tracking here, are you referring to the TSX data? Please clarify

--- *We used offset-tracking for both Sentinel-1 and TSX. At L260 we only describe the methods used in a more generalized form. What has finally been used for each sensor was described in L281, 287 and 294.*

L295: These dates are difficult to decipher. Do you mean '4 Nov. and 16 Nov., 2020'? If so, then write out the dates like this.

--- *Thank you for spotting, there are indeed two typos. It should read 'on 4. and 16.11.2020. For consistency with other text we have now changed this to 'on 4 and 16 November 2020*

L310-312: this statement requires better justification for ignoring microwave penetration in the SRTM DEM. Gardelle et al. (2012) state that it averages 3 m in the Karakoram, so this should be compared to the elevation changes caused by the surges.

--- *Considering the elevation changes caused by the surges (from -50 to +150 m), we think that a possible 3 m difference can be neglected, as it is not even systematic. Microwave penetration of the C band is highly depending on local conditions, in particular liquid water content, of the snow/ice during acquisition. These are not known for the SRTM DEM, which is a composite of several scenes acquired during February 2000. Assuming an elevation dependent correction, which is a common way of estimating SRTM penetration, would likely be a rough simplification that can be also criticised as inappropriate. As we simply do not know where to correct the SRTM elevations, we think it is better to just present the raw results. Given other uncertainties, also the impact on the calculated volume changes should be small. We have added a short explanation for this.*

L363: since you have information concerning the retreat rates of South Chongtar and NN9 prior to their recent surges (e.g., from Fig. S7), then I wonder if it would be useful to plot these on Fig. 4a? Seeing their rate of retreat would be useful to compare to their rate of advance, and help to support the retreat patterns that you describe later in this section.

--- *We have indeed also discussed presenting and including this information. However, for NN9 the number of usable satellite images is very small and the front was basically stationary from 1989 to 1998, meaning that any (quantitative) change assessment would be smaller than the uncertainty and thus of limited value. For South Chongtar, the stagnant post-surge ice mass is basically down-wasting in place from 1970 to 2000. As the lowermost part of the glacier (incl. its 'terminus') is increasingly debris covered, it is also difficult to identify its position. There is no clear indication where the terminus of the glacier is. After 2000 one can follow the change in the lowest position of the central clean-ice part, but this is not really the terminus as large amounts of ice are kept under the debris of lateral moraines. The observed clean-ice 'retreat' is only due to down-wasting of an otherwise stagnant ice mass rather than a dynamic reaction. We would thus argue that a comparison with the advance of the active phase might lead to wrong conclusions and prefer not showing it in the same plot. For the DEM differencing and volume change calculations we might have overestimated the volume loss part, but think that we cannot seriously quantify the impact (apart from a more theoretical test).*

L557: I'm unclear as to what 'compensation effects' is referring to here

--- *The 'compensation effect' is described in the next sentence. We have now placed a : after included to be clear that what follows is the explanation.*

L579: probably useful to remind the reader here and earlier in this section as to which sensors are optical (Landsat, Planet, Sentinel 2), and which are SAR (TSX, Sentinel 1).

*--- Agreed and added.*

L596: I'm unclear as to the date of the HMA 2015 DEM as the label at the top of Fig. S12 specifies 'spring 2015', but Table 2 indicates Feb.-Aug. 2015. Please clarify.
*--- Thank you for spotting this inconsistency. Indeed, the HMA DEM is a 7-month composite over this region but the majority of the region is covered by the images from April 2015. This is why we referred to it as spring in Fig. S6. However, we fully agree that this is a bit confusing and have thus removed the title from the image.*

L661: provide ref(s) to support the statement that these are likely the highest and lowest surge velocities in the Karakoram
*--- It is a bit of a speculation so we have written 'likely'. At L724-726 we cite some further studies reporting velocities on the high end. For the low end, the question is likely if the advance of North Chongtar at 20-100 m/y can be called a surge at all? Lv et al. (2020) suggested that the glacier with a near constant 20 m/y is not surging but just advancing, of course also noting that the typical elevation change pattern was absent. North Chongtar advanced over 40 years with a near constant 30 m/y rate and only showed acceleration and the typical mass relocation pattern afterwards. So from this perspective it would be classified as a surge-type glacier, but with a very slow advance rate. If the 40-year advance prior to the 'real' surge can also be named a surge is yet unclear. We would say yes but others might disagree.*

L674: clarify as to exactly what characteristics you're referring to when you say that these glaciers 'developed nearly all characteristics of a surge'
*--- Agreed and added: '(e.g. a heavily crevassed surface, shear margins, strong increase in flow velocity, high frontal advance rates, mass transfer from a reservoir to a receiving zone)'*

L697: can you refer to any modelled mass balance data from this region to indicate whether positive mass balance conditions have occurred there recently? That could help to support or refute your statements about potential causes of the observed glacier changes.
*--- We would likely not trust any modelled mass balance for this region, as we have no measured mass balances and don't know precipitation amounts. More trustworthy are geodetic measurements published earlier, but in a region dominated by surge-type glaciers they mostly show their mass transfer (see Fig 6d in the extended material by Hugonnet et al., 2021). It would also be difficult to conclude anything from the mass balance of a glacier nearby to conditions of other glaciers, as differences in topography / hypsometry might have a strong impact. Hence, we speculate here about the causes and have thus written 'might have'.*

L659-740: overall this section could do with better referencing to existing literature. For example, you refer several times to the characteristics of hydrologically controlled vs. thermally controlled surges, but don't refer to the previous literature which describes this (both for the Karakoram and elsewhere) and the evidence for the different types of flow patterns and lengths of active and quiescent phases. There are many 'classic' papers which can be useful here, such as Sevestre and Benn (2015), Murray et al. (2003), Benn et al. (2019; https://doi.org/10.1017/jog.2019.62), and Kamb et al. (1985; https://doi.org/10.1126/science.227.4686.469).
*--- We fully agree that more references can be added here and have done so based on your suggestions (and some more). Our intention to keep it a bit more basic and simple here was related to the fact that the interpretation of our observations is more speculative and different opinions on the theoretical foundations exist. Due to the diversity of the observed*

*changes we would also need to go through and basically review the existing theories. But as our study is already rather long and our interpretation might not be shared by other experts, we would prefer to keep the discussion concise and less detailed. As mentioned previously (e.g. Quincey et al. 2015), the so far used theories to explain surges might work well for the glaciers they have bee derived from, but in this region we just see a great mix of everything and have a hard time with convincing explanations.*

L748: it's confusing to have the (3) in brackets here; just write out the sentence to describe Sentinel-2 and Planet separately
*--- Agreed and done.*

L755: change 'relative' to 'relatively'
L758: change 'reliable' to 'reliably'
*--Both done.*

L765 & L810: a major difference between the sensors that you haven't mentioned is that some are optical and some are SAR, so it would be useful to discuss whether this has any impact on the measured velocities and how suitable different scene pairs are for deriving them. E.g., effect of snow cover, effect of surface melt, SAR layover/foreshortening effects, SAR penetration, optical shadows?
*--- The SAR data we could use for a comparison with optical sensors are the TerraSAR-X scenes we have discussed and presented in Figs. 5c, 6 and 7a. As described in the text, they match very well to the velocities derived from Landsat 8 data acquired in 2014. We have not used SAR summer image pairs but for the autumn / winter / spring scenes, results were very good. Limitations due to snow cover and shadow can be well seen in the 2D maps of Figs. 5 and 8 for the higher glacier parts of South Chongtar, but these are difficult to discern from limitations of spatial resolution for the two smaller glaciers. With Sentinel-2 we could also trace velocities from October 2020 to April 2021, as contrast was good (low solar elevation, limited snow, extreme crevassing). Cloud cover is the more important limitation during monsoon (May-August).*

L793: change word order to 'not yet'
*--- Done.*

L831-880: similar to my comment above for L659-740, the Conclusions would benefit from better referencing to previous studies so that it's clear how your findings compare to those of others. At the moment you don't have a single reference in this section!
*--- Thank you for the comment. Actually, the missing references in this section are fully on purpose as I (FP) have learned to not add anything new in the conclusions, but conclude what has been presented and discussed before. At least for the last 20 years this has never been criticised ;) As a reviewer, I would also always ask the authors to remove any citations from the conclusions section and present the related material in the discussion, where the place for arguments and references to other authors work should be. We acknowledge your different view on this but, if you don't mind, would prefer to keep the conclusions free of citations.*

L875: would be more accurate to say: '…the termini of South Chongtar and NN9 are now colliding"
*--- Yes, agreed and changed.*

**Tables and Figures**

Table 3: Add reference to Fig. S5 in the caption so that it's clear as to which areas were used for the volume calculations

*--- Fully agreed, added.*

Fig. 1: I would like to see distance markers along the centrelines, so that it's easier to understand how the distances shown in other figures (e.g., Figs. 6, 7) relate to physical locations along the glaciers

*--- Yes, good idea. Thank you for the suggestion.*

Fig. 2: It would be useful to split the lines into two groupings: those that rely on DEMs (upper), and those that rely on optical or SAR images (bottom), such as by adding a blank line between them and changing the 'Sensors' label on the y-axis to something more descriptive. This is because some of the DEM sources (e.g., ASTER, SPOT) can also provide optical images, but you don't use images of that type in this study.

*--- Agreed, this would bring extra clarity. We have currently used different colours (blue and orange lines, green circles) to differentiate the datasets, but this is only explained in the caption rather than in a legend. So we have now added a legend, inserted some space and revised the axes.*

Fig. 4: increase the font size for the legend

*--- Done.*

Figs. 5, 8, 11: the velocities would be easier to see if you cropped out the data to only show it over glaciers, and not over bedrock

*--- This is agreed but we would lose the possibility to show that velocities and elevation changes off glaciers are really close to zero. As a compromise, we have now prepared Figures with thicker outlines for the investigated glaciers (Fig. 11) and a transparency of 0.4 for the colour map off glacier (Figs. 5 and 8).*

Fig. 6: I find it hard to understand the dates of some of these velocity profiles as the colours are pretty similar in the legend (the Landsats are all a pretty similar blue, the more recent S2s all a similar red). Can you use a better colour scale to help separate them, or perhaps used dashed lines for some periods?

*--- Agreed, the colours are a bit close. The colour scale for this and the other figures is optimized for colour-blind people so the number of shades is given. But in particular for Fig. 6 lines are difficult to identify. We will experiment with your suggestion using dashed lines or annotations within the figure.*

Fig. 9: Add a reference to Fig. 1 in the caption to indicate where the cross-profile is located. Also misspelling of 'Grey numbers…'

*--- Done.*

**Supplemental**

L12: Sentine-2 is misspelled

Table S1: please use the format yyyy/mm/dd for the dates so that they're consistent with Table 2. Also use the same font size throughout.

Fig. S2 and others: state in the caption as to what coordinate system is used for these plots (e.g., UTM 34N?)

*--- Thank you for spotting, all done.*

---

## Author Comment (AC2)

**Response to the reviewers comments**

**RC2: 'Comment on tc-2021-370', Anonymous Referee #2**

**General comments**

Authors report on the surge behaviour of three glaciers in the same valley discussing differences in timing, extent and progression of the surges. For this purpose, they use a comprehensive dataset comprising different optical and radar satellites to map glacier advance, flow velocities and elevation changes.

The paper is well written with high quality figures and it also gives a good overview on the literature about glacier surges.

There is only one aspect that should be elaborated more carefully: In addition to the analysis of the surge behaviour of the investigated glaciers, authors compare results from different sensors in order to discuss their suitability for similar studies. This intercomparison has to be done in a more quantitative way. Statements like "average differences are insignificant" or "agree very well" are not enough in this context and error measures should be given (see specific comments).

I suggest to accept the manuscript after minor revisions.

*--- Thank you very much for the careful reading and constructive comments! We will add some quantitative numbers to the various dataset inter-comparisons we have performed.*

**Specific comments**

line 212 Could you please briefly state what kind of DEM datasets?

*--- It just means here that the HMA DEM is a composite of datasets predominantly acquired at different times within 2015 that cover our study region. All studied glacier tongues are predominantly based on data from 2015 (eight scenes from Feb.-Aug. 2015, two scenes from October 2015 over parts of accumulation areas), complemented with four scenes from 2009-2011 in the very East of South/North Chongtar accumulation areas, and a thin stripe from 2009 covering parts of the South Chongtar/Sarpo Laggo glacier tongues. They have all been derived from the same very high-resolution sensors (WorldView 2/3, GeoEye 1) as described by Shean et al. (2016). We can obtain the acquisitions dates from the image footprints, but we do not know by which technique the datasets have been merged where footprints overlap. We have specified the DEM origin within the text and added detailed information about the contributing datasets to Fig. S12.*

line 507 "some surface lowering": How much?

*--- It is about 10-15 m (added).*

line 562 Which area was used to calculate the volume loss of Sarpo Laggo Glacier? In Fig. 11 a) and b) there is volume gain visible in the part of the glacier that is not covered in Fig. 11 c and d.

*--- Yes, agreed. We have only calculated the loss part of the volume change (marked in Fig. S5) as we were interested in how this loss compares to the total gain of all other (much smaller) glaciers. Indeed, there is quite some elevation gain higher up on Sarpo Laggo (also visible in Fig. S13f), also for some of the upper tributary glaciers. This is why we have written 'the (uncompensated) volume loss', meaning we have neglected here the volume gain part.*

Section 5.6 Uncertainties should be placed before 5.5 Sensor intercomparison. As a result, the differences between the sensors can be quantitatively compared to the determined uncertainties.
*--- Yes, this is possible and has been changed. A comparison of sensor differences with uncertainties has been added.*

line 572 Give error measures and compare with the sensor uncertainties derived in 5.6.
*--- Done.*

line 597 "agree very well": Compare it quantitatively with the stable terrain accuracies (Table S3).
*--- Done.*

line 630 MAD: Table S2 only lists mean and std. Either explain how you derive std from MAD or adjust text and table. Check all numbers cited in 5.6.2 accordingly.
*--- MAD values have been added.*

line 667 Explain why "maximum flow velocities in summer" are atypical.
*--- They are not atypical for 'usual' glaciers as the higher lubrication by meltwater reaching the bed causes more basal sliding and thus higher velocities. However, they are atypical for this specific surge mechanism (controlled by changes in basal hydrology) as the switch from an efficient to an unefficient basal drainage system usually only occurs during winter. We have now added a related explanation.*

line 740 Can you comment on the influence of crevassing on the volume determination? Are the crevasses resolved by the DEMs? Does the DEM of the crevassed surface represent a mean elevation or is it more like an envelope curve?
*--- As shown by the DEM hillshades in Fig. S1, the HMA DEM and the SPOT DEM from 2020 both see crevasses, at least the upper part of the big ones: The crevasses/bulges on South Chongtar around 3 973 000 N / 620 500 E measure ca. 50 m across and are clearly resolved by the higher-resolution DEM, but averaged out by the 30 m-DEMs Fig. S6f, shows profiles across the down-wasting Sarppo Lago tongue before arrival of the South Chongtar surge bulge, i.e., surface features are very similiar for a year or more. It indicates that the SPOT2020 DEM resolves larger surface structures (ca. 50 m across at 3978 km N) nearly as well as ICESat-2, albeit some of the depth is lost, but ignores smaller crevasses (ca. 10 m across at 3,977 km N), and in that case they are treated like an envelope curve. In the same panel, it can be seen that the 30 m SPOT2015 DEM represents an average of the surface captured by the 5 m HMA DEM. In conclusion, the DEMs present a mean elevation of the surface and only the largest crevasses might have been considered for the volume change. The uncertainty in the resulting volume change is thus likely mostly impacted by the uncertainties of the DEMs and the influence of crevassing is small.*

line 770 basically identical: But they are from different seasons and Fig. S12 shows the differences. I suggest to give median and MAD to quantify this statement.
*--- Median elevation differences off glacier (see Table S3) are -0.11 m (compared to minimum / maximum differences of -51 and +423 m found at artifacts), which we think can be named 'basically identical'. From the difference image in Fig. S12 we would not conclude that seasonal changes can be seen over glacier surfaces (apart from the advancing North Chongtar terminus), but there may be some influence from contributing DEM data in some parts. We clarified this in the text and added further details to the caption of Fig. S12.*

line 858 Was Sentinel-1 used in Section 5.1 or only in the animation? Sentinel-1 images are not listed in Table S1.
--- *We originally had the intention using Sentinel-1 to derive velocity fields, but after this didn't work we only did a few tests shown in Fig. S4. The scene pair is thus not listed in Table S2 but only in L295 (with some typos). The animation in the Supplement is just for illustration purposes.*

**Technical corrections**
line 27 same region: I'd narrow it down to the same valley, even.
--- *As a valley can be very long, this would not be more restrictive. We have now written 'small region'.*

line 43 Is it ICESat-2 ATL03?
--- *Yes, indeed. Thank you for spotting!*

line 102 by Hugonnet et al. (2021): Please make it clearer that this is an external dataset.
--- *Done.*

line 107 35.94°N
--- *Done.*

line 139 At their historically recorded maximum extent
--- *Done.*

line 181 Sentinel-1 is not included in table S1.
--- *This is because Sentienl-1 has not been used to derive geometric changes.*

line 295 Correct date format.
--- *Done.*

line 299 Which standard processing steps? If these are listed in the following passage, use a colon (:) at the end of this phrase.
--- *Done.*

line 305 "We also excluded data voids": Seems clear. How could you alternatively include voids?
--- *Fully agreed, this is a bit misleading and has been rewritten.*

line 438 Replace "afterwards" with "further down".
--- *Done.*

line 450 Fig. 7a confirms this observation:
--- *Done.*

line 451 "(up to 0.2 m d-1)": Two lines above the maximum is 0.4 m/d.
--- *Agreed. We include up to 0.4 m/d for the pre-surge phase and would argue that afterwards the surge took off.*

line 472 Replace "3 km" with "km 3" like in line 474
--- *Done. We were not sure if this is grammatically possible at all.*

line 474 What do you mean with "maximum velocities near km 15 changed only slowly over the entire glacier length"? Do you refer to the location km 15 or the entire length?
*--- Agreed, this is confusing. We have now written: 'maximum velocities are found near km 15 and decreased only slowly at this location and over large parts of the glacier length (back to km 6) at about the same rate.'*

line 475 Replace "10 km" with "km 10".
*--- Done.*

lines 590, 592, 597: Specify subfigure in Fig. S6 (see comment on S6 below).
*--- Done, thank you for spotting.*

line 643 median elevation differences to the reference DEM "on bedrock"
*--- Done.*

line 670 Replace "implies" with "imply".
*--- Done.*

line 708 "location of the maximum away from the centre": Do you mean centre line?
*--- Yes, changed.*

line 764 strictly or strict?
*--- Strict sounds better, changed.*

line 799 short-term variation
*--- Done.*

line 834 in the same valley?
*--- As above, we have no written small region.*

line 836 Also give the duration of the active phase in comparison to North Chongtar.
*--- Done for both.*

Table 3, line 1150 numbers
*--- Done.*

Figure 1 Add map grid
*--- We added a coordinate cross with its latitude and longitude listed in the caption.*

In some figures (e.g. Fig. 6, 7) colours are hard to distinguish. Please check whether you could use colours that are better distinguishable.
*--- Agreed, in particular for Fig. 6 it is a bit difficult. For Fig. 7a/b we think it is still possible as there is a clear increasing / decreasing pattern that can be linked to the temporal evolution. As this is a colour palette for colour blind people that we wanted to keep. The revised version of the figures (6a, 6b and 7a) include different symbols to better distinguish the colors.*

Figure 9 State direction of cross profile. West to east?
*--- It is SW to NE (added).*

line 1219 with respect to
*--- Done.*

**Supplemental Material**

Table S1: Add Sentinel-1.

*--- Not done (as it Sentinel-1 has not been used for delineating glacier extents or determination of frontal advance rates.*

Table S3: Why are some lines in italics? I suggest to add NMAD.

*Statistics in regular font are accuracy measures from differences with regard to the HMA DEM (co-registration base). Those in italic font are statistics for other computed DEM differences, i.e. not with regard to the HMA DEM and thus not "official" accuracy measures, but may be of interest nevertheless. An explanation has been added to the table caption. We computed the NMAD and added the values to the table and text.*

Fig. S1: Check dates, resolutions and dataset names as they are inconsistent with Table 2. Subfigure d) Use the same clipping as in a)-c).

*--- These are subsets of the original DEMs, i.e. at their native resolution. For this reason spatial resolution is different from the resampled DEMs used to calculate elevation changes. The clipping has been adjusted for panel d.*

Fig. S5, line 60: What are "biased accumulation areas"?

*--- Changed to '… to remove no data or accumulation areas biased by artifacts.'*

Fig. S6: Consider numbering all panels as subfigures. It makes it easier to refer to a specific panel in the text.

*--- Done.*

line 66 2 February 2019: Does not coincide with date on map. Shouldn't it be December?

*--- Yes, December is correct. Thanks for spotting.*

Fig. S7: clockwise? The main paper has the usual order from left to right (e.g. Fig. 5). Please adjust.

*--- Yes, the clockwise here is on purpose.*

Fig. S13: Dates (and order of dates) do not coincide with the ones in the figure for a) to d).

*--- Yes, agreed. This has been adjusted.*